# Asymmetric Certified Robustness via Feature-Convex Neural Networks

**Samuel Pfrommer**[*]     **Brendon G. Anderson**[*]     **Julien Piet**     **Somayeh Sojoudi**

University of California, Berkeley

{sam.pfrommer,bganderson,piet,sojoudi}@berkeley.edu

## Abstract

Real-world adversarial attacks on machine learning models often feature an asymmetric structure wherein adversaries only attempt to induce false negatives (e.g., classify a spam email as not spam). We formalize the asymmetric robustness certification problem and correspondingly present the *feature-convex neural network* architecture, which composes an input-convex neural network (ICNN) with a Lipschitz continuous feature map in order to achieve asymmetric adversarial robustness. We consider the aforementioned binary setting with one "sensitive" class, and for this class we prove deterministic, closed-form, and easily-computable certified robust radii for arbitrary $\ell_p$-norms. We theoretically justify the use of these models by extending the universal approximation theorem for ICNN regression to the classification setting, and proving a lower bound on the probability that such models perfectly fit even unstructured uniformly distributed data in sufficiently high dimensions. Experiments on Malimg malware classification and subsets of the MNIST, Fashion-MNIST, and CIFAR-10 datasets show that feature-convex classifiers attain substantial certified $\ell_1$, $\ell_2$, and $\ell_\infty$-radii while being far more computationally efficient than competitive baselines. [1]

## 1 Introduction

Although neural networks achieve state-of-the-art performance across a range of machine learning tasks, researchers have shown that they can be highly sensitive to adversarial inputs that are maliciously designed to fool the model [11, 61, 52]. For example, the works Eykholt et al. [22] and Liu et al. [42] show that small physical and digital alterations of vehicle traffic signs can cause image classifiers to fail. In safety-critical applications of neural networks, such as autonomous driving [12, 69] and medical diagnostics [1, 71], this sensitivity to adversarial inputs is clearly unacceptable.

A line of heuristic defenses against adversarial inputs has been proposed, only to be defeated by stronger attack methods [14, 36, 7, 64, 46]. This has led researchers to develop certifiably robust methods that provide a provable guarantee of safe performance. The strength of such certificates can be highly dependent on network architecture; general off-the-shelf models tend to have large Lipschitz constants, leading to loose Lipschitz-based robustness guarantees [29, 23, 73]. Consequently, lines of work that impose certificate-amenable structures onto networks have been popularized, e.g., specialized model layers [63, 77], randomized smoothing-based networks [41, 18, 76, 72, 3], and ReLU networks that are certified using convex optimization and mixed-integer programming [68, 67, 54, 4, 45]. The first category only directly certifies against one specific choice of norm, producing poorly scaled radii for other norms in high dimensions. The latter two approaches incur serious computational challenges: randomized smoothing typically requires the classification of thousands

---

[*]Equal contribution.

[1]Code for reproducing our results is available on GitHub.

of randomly perturbed samples per input, while optimization-based solutions scale poorly to large networks.

Despite the moderate success of these certifiable classifiers, conventional assumptions in the literature are unnecessarily restrictive for many practical adversarial settings. Specifically, most works consider a multiclass setting where certificates are desired for inputs of any class. By contrast, many real-world adversarial attacks involve a binary setting with only one *sensitive class* that must be robust to adversarial perturbations. Consider the representative problem of spam classification; a malicious adversary crafting a spam email will only attempt to fool the classifier toward the "not-spam" class— never conversely [20]. Similar logic applies for a range of applications such as malware detection [28], malicious network traffic filtering [56], fake news and social media bot detection [19], hate speech removal [27], insurance claims filtering [24], and financial fraud detection [15].

The important asymmetric nature of these classification problems has long been recognized in various subfields, and some domain-specific attempts at robustification have been proposed with this in mind. This commonly involves robustifying against adversaries appending features to the classifier input. In spam classification, such an attack is known as the "good word" attack [44]. In malware detection, numerous approaches have been proposed to provably counter such additive-only adversaries using special classifier structures such as non-negative networks [25] and monotonic classifiers [32]. We note these works strictly focus on *additive* adversaries and cannot handle general adversarial perturbations of the input that are capable of perturbing existing features. We propose adding this important asymmetric structure to the study of norm ball-certifiably robust classifiers. This narrowing of the problem to the asymmetric setting provides prospects for novel certifiable architectures, and we present feature-convex neural networks as one such possibility.

## 1.1 Problem Statement and Contributions

This section formalizes the *asymmetric robustness certification problem* for general norm-bounded adversaries. Specifically, we assume a binary classification setting wherein one class is "sensitive"— meaning we seek to certify that, if some input is classified into this sensitive class, then adversarial perturbations of sufficiently small magnitude cannot change the prediction.

Formally, consider a binary classifier $f_\tau \colon \mathbb{R}^d \to \{1, 2\}$, where class 1 is the sensitive class for which we desire certificates. We take $f_\tau$ to be a standard thresholded version of a soft classifier $g \colon \mathbb{R}^d \to \mathbb{R}$, expressible as $f_\tau(x) = T_\tau(g(x))$, where $T_\tau \colon \mathbb{R} \to \{1, 2\}$ is the thresholding function defined by

$$T_\tau(y) = \begin{cases} 1 & \text{if } y + \tau > 0, \\ 2 & \text{if } y + \tau \le 0, \end{cases} \tag{1}$$

with $\tau \in \mathbb{R}$ being a user-specified parameter that shifts the classification threshold. A classifier $f_\tau$ is considered certifiably robust at a class 1 input $x \in \mathbb{R}^d$ with a radius $r(x) \in \mathbb{R}_+$ if $f_\tau(x + \delta) = f_\tau(x) = 1$ for all $\delta \in \mathbb{R}^d$ with $\|\delta\| < r(x)$ for some norm $\|\cdot\|$. Thus, $\tau$ induces a tradeoff between the clean accuracy on class 2 and certification performance on class 1. As $\tau \to \infty$, $f_\tau$ approaches a constant classifier which achieves infinite class 1 certified radii but has zero class 2 accuracy.

For a particular choice of $\tau$, the performance of $f_\tau$ can be analyzed similarly to a typical certified classifier. Namely, it exhibits a class 2 clean accuracy $\alpha_2(\tau) \in [0, 1]$ as well as a class 1 certified accuracy surface $\Gamma$ with values $\Gamma(r, \tau) \in [0, 1]$ that capture the fraction of the class 1 samples that can be certifiably classified by $f_\tau$ at radius $r \in \mathbb{R}_+$. The class 1 clean accuracy $\alpha_1(\tau) = \Gamma(0, \tau)$ is inferable from $\Gamma$ as the certified accuracy at $r = 0$.

The full asymmetric certification performance of the family of classifiers $f_\tau$ can be captured by plotting the surface $\Gamma(r, \tau)$, as will be shown in Figure 1a. Instead of plotting against $\tau$ directly, we plot against the more informative difference in clean accuracies $\alpha_1(\tau) - \alpha_2(\tau)$. This surface can be viewed as an asymmetric robustness analogue to the classic receiver operating characteristic curve.

Note that while computing the asymmetric robustness surface is possible for our feature-convex architecture (to be defined shortly), it is computationally prohibitive for conventional certification methods. We therefore standardize our comparisons throughout this work to the certified accuracy cross section $\Gamma(r, \tau^*)$ for a $\tau^*$ such that clean accuracies are balanced in the sense that $\alpha_2(\tau^*) = \alpha_1(\tau^*)$, noting that $\alpha_1$ monotonically increases in $\tau$ and $\alpha_2$ mononically decreases in $\tau$. We discuss finding such a $\tau^*$ in Appendix E.4. This choice allows for a direct comparison of the resulting certified accuracy curves without considering the non-sensitive class clean accuracy.

With the above formalization in place, the goal at hand is two-fold: 1) develop a classification architecture tailored for the asymmetric setting with high robustness, as characterized by the surface $\Gamma$, and 2) provide efficient methods for computing the certified robust radii $r(x)$ used to generate $\Gamma$.

**Contributions.** We tackle the above two goals by proposing *feature-convex neural networks* and achieve the following contributions:

1. We exploit the feature-convex structure of the proposed classifier to provide asymmetrically tailored closed-form class 1 certified robust radii for arbitrary $\ell_p$-norms, solving the second goal above and yielding efficient computation of $\Gamma$.

2. We characterize the decision region geometry of convex classifiers, extend the universal approximation theorem for input-convex $\mathrm{ReLU}$ neural networks to the classification setting, and show that convex classifiers are sufficiently expressive for high-dimensional data.

3. We evaluate against several baselines on MNIST 3-8 [37], Malimg malware classification [50], Fashion-MNIST shirts [70], and CIFAR-10 cats-dogs [35], and show that our classifiers yield certified robust radii competitive with the state-of-the-art, empirically addressing the first goal listed above.

All proofs and appendices can be found in the Supplemental Material.

## 1.2 Related Works

**Certified adversarial robustness.** Three of the most popular approaches for generating robustness certificates are Lipschitz-based bounds, randomized smoothing, and optimization-based methods. Successfully bounding the Lipschitz constant of a neural network can give rise to an efficient certified radius of robustness, e.g., via the methods proposed in Hein and Andriushchenko [29]. However, in practice such Lipschitz constants are too large to yield meaningful certificates, or it is computationally burdensome to compute or bound the Lipschitz constants in the first place [65, 23, 73]. To overcome these computational limitations, certain methods impose special structures on their model layers to provide immediate Lipschitz guarantees. Specifically, Trockman and Kolter [63] uses the Cayley transform to derive convolutional layers with immediate $\ell_2$-Lipschitz constants, and Zhang et al. [77] introduces a $\ell_\infty$-distance neuron that provides similar Lipschitz guarantees with respect to the $\ell_\infty$-norm. We compare with both these approaches in our experiments.

Randomized smoothing, popularized by Lecuyer et al. [38], Li et al. [41], Cohen et al. [18], uses the expected prediction of a model when subjected to Gaussian input noise. These works derive $\ell_2$-norm balls around inputs on which the smoothed classifier remains constant, but suffer from nondeterminism and high computational burden. Follow-up works generalize randomized smoothing to certify input regions defined by different metrics, e.g., Wasserstein, $\ell_1$-, and $\ell_\infty$-norms [39, 62, 72]. Other works focus on enlarging the certified regions by optimizing the smoothing distribution [76, 21, 5], incorporating adversarial training into the base classifier [57, 78], and employing dimensionality reduction at the input [53].

Optimization-based certificates typically seek to derive a tractable over-approximation of the set of possible outputs when the input is subject to adversarial perturbations, and show that this over-approximation is safe. Various over-approximations have been proposed, e.g., based on linear programming and bounding [68, 67], semidefinite programming [54], and branch-and-bound [4, 45, 66]. The $\alpha, \beta$-CROWN method [66] uses an efficient bound propagation to linearly bound the neural network output in conjunction with a per-neuron branching heuristic to achieve state-of-the-art certified radii, winning both the 2021 and the 2022 VNN certification competitions [8, 48]. In contrast to optimization-based methods, our approach directly exploits the convex structure of input-convex neural networks to derive closed-form robustness certificates, altogether avoiding any efficiency-tightness tradeoffs.

**Input-convex neural networks.** Input-convex neural networks, popularized by Amos et al. [2], are a class of parameterized models whose input-output mapping is convex. The authors develop tractable methods to learn input-convex neural networks, and show that such models yield state-of-the-art results in a variety of domains where convexity may be exploited, e.g., optimization-based inference. Subsequent works propose novel applications of input-convex neural networks in areas

such as optimal control and reinforcement learning [16, 75], optimal transport [47], and optimal power flow [17, 79]. Other works have generalized input-convex networks to input-invex networks [58, 51] and global optimization networks [80] so as to maintain the benign optimization properties of input-convexity. The authors of Siahkamari et al. [59] present algorithms for efficiently learning convex functions, while Chen et al. [16], Kim and Kim [34] derive universal approximation theorems for input-convex neural networks in the convex regression setting. The work Sivaprasad et al. [60] shows that input-convex neural networks do not suffer from overfitting, and generalize better than multilayer perceptrons on common benchmark datasets. In this work, we incorporate input-convex neural networks as a part of our feature-convex architecture and leverage convexity properties to derive novel robustness guarantees.

## 1.3 Notations

The sets of natural numbers, real numbers, and nonnegative real numbers are denoted by $\mathbb{N}$, $\mathbb{R}$, and $\mathbb{R}_+$ respectively. The $d \times d$ identity matrix is written as $I_d \in \mathbb{R}^{d \times d}$, and the identity map on $\mathbb{R}^d$ is denoted by $\mathrm{Id} \colon x \mapsto x$. For $A \in \mathbb{R}^{n \times d}$, we define $|A| \in \mathbb{R}^{n \times d}$ by $|A|_{ij} = |A_{ij}|$ for all $i, j$, and we write $A \geq 0$ if and only if $A_{ij} \geq 0$ for all $i, j$. The $\ell_p$-norm on $\mathbb{R}^d$ is given by $\| \cdot \|_p \colon x \mapsto (|x_1|^p + \cdots + |x_d|^p)^{1/p}$ for $p \in [1, \infty)$ and by $\| \cdot \|_p \colon x \mapsto \max\{|x_1|, \ldots, |x_d|\}$ for $p = \infty$. The dual norm of $\| \cdot \|_p$ is denoted by $\| \cdot \|_{p,*}$. The convex hull of a set $X \subseteq \mathbb{R}^d$ is denoted by $\mathrm{conv}(X)$. The subdifferential of a convex function $g \colon \mathbb{R}^d \to \mathbb{R}$ at $x \in \mathbb{R}^d$ is denoted by $\partial g(x)$. If $\epsilon \colon \Omega \to \mathbb{R}^d$ is a random variable on a probability space $(\Omega, \mathcal{B}, \mathbb{P})$ and $P$ is a predicate defined on $\mathbb{R}^d$, then we write $\mathbb{P}(P(\epsilon))$ to mean $\mathbb{P}(\{\omega \in \Omega : P(\epsilon(\omega))\})$. Lebesgue measure on $\mathbb{R}^d$ is denoted by $m$. We define $\mathrm{ReLU} \colon \mathbb{R} \to \mathbb{R}$ as $\mathrm{ReLU}(x) = \max\{0, x\}$, and if $x \in \mathbb{R}^d$, $\mathrm{ReLU}(x)$ denotes $(\mathrm{ReLU}(x_1), \ldots, \mathrm{ReLU}(x_d))$. We recall the threshold function $T_\tau \colon \mathbb{R} \to \{1, 2\}$ defined by (1), and we define $T = T_0$. For a function $\varphi \colon \mathbb{R}^d \to \mathbb{R}^q$ and $p \in [1, \infty]$, we define $\mathrm{Lip}_p(\varphi) = \inf\{K \geq 0 : \|\varphi(x) - \varphi(x')\|_p \leq K\|x - x'\|_p \text{ for all } x, x' \in \mathbb{R}^d\}$, and if $\mathrm{Lip}_p(\varphi) < \infty$ we say that $\varphi$ is Lipschitz continuous with constant $\mathrm{Lip}_p(\varphi)$ (with respect to the $\ell_p$-norm).

## 2 Feature-Convex Classifiers

Let $d, q \in \mathbb{N}$ and $p \in [1, \infty]$ be fixed, and consider the task of classifying inputs from a subset of $\mathbb{R}^d$ into a fixed set of classes $\mathcal{Y} \subseteq \mathbb{N}$. In what follows, we restrict to the binary setting where $\mathcal{Y} = \{1, 2\}$ and class 1 is the sensitive class for which we desire robustness certificates (Section 1). In Appendix A, we briefly discuss avenues to generalize our framework to multiclass settings using one-versus-all and sequential classification methodologies and provide a proof-of-concept example for the Malimg dataset.

We now formally define the classifiers considered in this work. Note that the classification threshold $\tau$ discussed in Section 1.1 is omitted for simplicity.

**Definition 2.1.** Let $f \colon \mathbb{R}^d \to \{1, 2\}$ be defined by $f(x) = T(g(\varphi(x)))$ for some $\varphi \colon \mathbb{R}^d \to \mathbb{R}^q$ and some $g \colon \mathbb{R}^q \to \mathbb{R}$. Then $f$ is said to be a *feature-convex classifier* if the *feature map* $\varphi$ is Lipschitz continuous with constant $\mathrm{Lip}_p(\varphi) < \infty$ and $g$ is a convex function.

We denote the class of all feature-convex classifiers by $\mathcal{F}$. Furthermore, for $q = d$, the subclass of all feature-convex classifiers with $\varphi = \mathrm{Id}$ is denoted by $\mathcal{F}_{\mathrm{Id}}$.

As we will see in Section 3.1, defining our classifiers using the composition of a convex classifier with a Lipschitz feature map enables the fast computation of certified regions in the input space. This naturally arises from the global underestimation of convex functions by first-order Taylor approximations. Since sublevel sets of such $g$ are restricted to be convex, the feature map $\varphi$ is included to increase the representation power of our architecture (see Appendix B for a motivating example). In practice, we find that it suffices to choose $\varphi$ to be a simple map with a small closed-form Lipschitz constant. For example, in our experiments that follow with $q = 2d$, we choose $\varphi(x) = (x - \mu, |x - \mu|)$ with a constant channel-wise dataset mean $\mu$, yielding $\mathrm{Lip}_1(\varphi) \leq 2$, $\mathrm{Lip}_2(\varphi) \leq \sqrt{2}$, and $\mathrm{Lip}_\infty(\varphi) \leq 1$. Although this particular choice of $\varphi$ is convex, the function $g$ need not be monotone, and therefore the composition $g \circ \varphi$ is nonconvex in general. The prediction and certification of feature-convex classifiers are illustrated in Figure 1b.

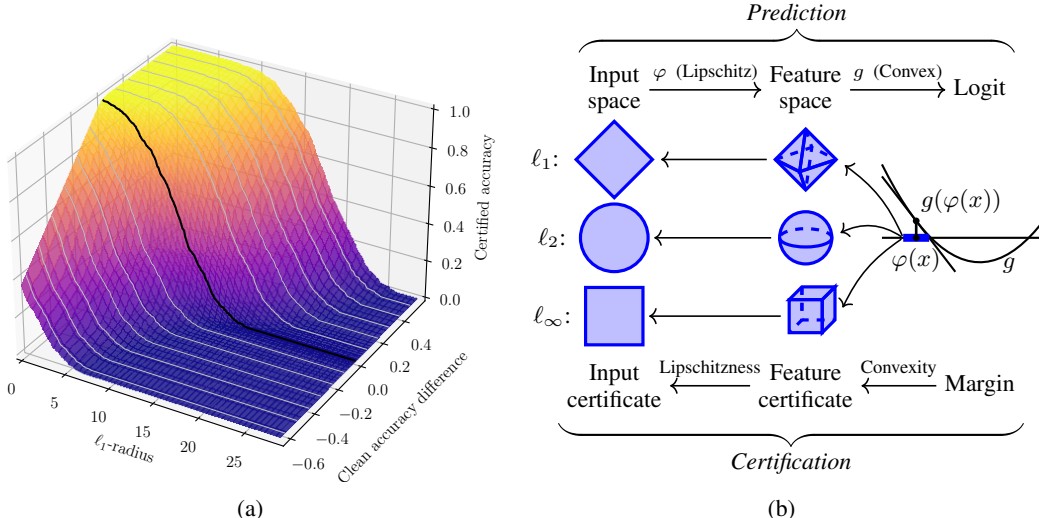

(a)                                      (b)

Figure 1: (a) The asymmetric certified accuracy surface $\Gamma(r, \tau)$ for MNIST 3-8, as described in Section 1.1. The "clean accuracy difference" axis plots $\alpha_1(\tau) - \alpha_2(\tau)$, and the black line highlights the certified robustness curve for when clean accuracy is equal across the two classes. (b) Illustration of feature-convex classifiers and their certification. Since $g$ is convex, it is globally underapproximated by its tangent plane at $\varphi(x)$, yielding certified sets for norm balls in the higher-dimensional feature space. Lipschitzness of $\varphi$ then yields appropriately scaled certificates in the original input space.

In practice, we implement feature-convex classifiers using parameterizations of $g$, which we now make explicit. Following Amos et al. [2], we instantiate $g$ as a neural network with nonnegative weight matrices and nondecreasing convex nonlinearities. Specifically, we consider ReLU nonlinearities, which is not restrictive, as our universal approximation result in Theorem 3.6 proves.

**Definition 2.2.** A *feature-convex* ReLU *neural network* is a function $\hat{f} \colon \mathbb{R}^d \to \{1, 2\}$ defined by $\hat{f}(x) = T(\hat{g}(\varphi(x)))$ with $\varphi \colon \mathbb{R}^d \to \mathbb{R}^q$ Lipschitz continuous with constant $\mathrm{Lip}_p(\varphi) < \infty$ and $\hat{g} \colon \mathbb{R}^q \to \mathbb{R}$ defined by

$$\hat{g}(x^{(0)}) = A^{(L)}x^{(L-1)} + b^{(L)} + C^{(L)}x^{(0)}, \quad x^{(l)} = \mathrm{ReLU}\left(A^{(l)}x^{(l-1)} + b^{(l)} + C^{(l)}x^{(0)}\right),$$

for all $l \in \{1, 2, \ldots, L-1\}$ for some $L \in \mathbb{N}$, $L > 1$, and for some consistently sized matrices $A^{(l)}, C^{(l)}$ and vectors $b^{(l)}$ satisfying $A^{(l)} \geq 0$ for all $l \in \{2, 3, \ldots, L\}$.

Going forward, we denote the class of all feature-convex ReLU neural networks by $\hat{\mathcal{F}}$. Furthermore, if $q = d$, the subclass of all feature-convex ReLU neural networks with $\varphi = \mathrm{Id}$ is denoted by $\hat{\mathcal{F}}_{\mathrm{Id}}$, which corresponds to the input-convex ReLU neural networks proposed in Amos et al. [2].

For every $\hat{f} \in \hat{\mathcal{F}}$, it holds that $\hat{g}$ is convex due to the rules for composition and nonnegatively weighted sums of convex functions [13, Section 3.2], and therefore $\hat{\mathcal{F}} \subseteq \mathcal{F}$ and $\hat{\mathcal{F}}_{\mathrm{Id}} \subseteq \mathcal{F}_{\mathrm{Id}}$. The "passthrough" weights $C^{(l)}$ were originally included by Amos et al. [2] to improve the practical performance of the architecture. In some of our more challenging experiments that follow, we remove these passthrough operations and instead add residual identity mappings between hidden layers, which also preserves convexity. We note that the transformations defined by $A^{(l)}$ and $C^{(l)}$ can be taken to be convolutions, which are nonnegatively weighted linear operations and thus preserve convexity [2].

## 3   Certification and Analysis of Feature-Convex Classifiers

We present our main theoretical results in this section. First, we derive asymmetric robustness certificates (Theorem 3.1) for our feature-convex classifiers in Section 3.1. Then, in Section 3.2, we introduce the notion of convexly separable sets in order to theoretically characterize the representation power of our classifiers. Our primary representation results give a universal function approximation

theorem for our classifiers with $\varphi = \mathrm{Id}$ and $\mathrm{ReLU}$ activation functions (Theorem 3.6) and show that such classifiers can perfectly fit convexly separable datasets (Theorem 3.7), including the CIFAR-10 cats-dogs training data (Fact 3.8). We also show that this strong learning capacity generalizes by proving that feature-convex classifiers can perfectly fit high-dimensional uniformly distributed data with high probability (Theorem 3.10).

## 3.1 Certified Robustness Guarantees

In this section, we address the asymmetric certified robustness problem by providing class 1 robustness certificates for feature-convex classifiers $f \in \mathcal{F}$. Such robustness corresponds to proving the absence of false negatives in the case that class 1 represents positives and class 2 represents negatives. For example, if in a malware detection setting class 1 represents malware and class 2 represents non-malware, the following certificate gives a lower bound on the magnitude of the malware file alteration needed in order to misclassify the file as non-malware.

**Theorem 3.1.** *Let $f \in \mathcal{F}$ be as in Definition 2.1 and let $x \in f^{-1}(\{1\}) = \{x' \in \mathbb{R}^d : f(x') = 1\}$. If $\nabla g(\varphi(x)) \in \mathbb{R}^q$ is a nonzero subgradient of the convex function $g$ at $\varphi(x)$, then $f(x + \delta) = 1$ for all $\delta \in \mathbb{R}^d$ such that*

$$\|\delta\|_p < r(x) := \frac{g(\varphi(x))}{\mathrm{Lip}_p(\varphi)\|\nabla g(\varphi(x))\|_{p,*}}.$$

*Remark* 3.2. For $f \in \mathcal{F}$ and $x \in f^{-1}(\{1\})$, a subgradient $\nabla g(\varphi(x)) \in \mathbb{R}^q$ of $g$ always exists at $\varphi(x)$, since the subdifferential $\partial g(\varphi(x))$ is a nonempty closed bounded convex set, as $g$ is a finite convex function on all of $\mathbb{R}^q$—see Theorem 23.4 in Rockafellar [55] and the discussion thereafter. Furthermore, if $f$ is not a constant classifier, such a subgradient $\nabla g(\varphi(x))$ must necessarily be nonzero, since, if it were zero, then $g(y) \geq g(\varphi(x)) + \nabla g(\varphi(x))^\top (y - \varphi(x)) = g(\varphi(x)) > 0$ for all $y \in \mathbb{R}^q$, implying that $f$ identically predicts class 1, which is a contradiction. Thus, the certified radius given in Theorem 3.1 is always well-defined in practical settings.

Theorem 3.1 is derived from the fact that a convex function is globally underapproximated by any tangent plane. The nonconstant terms in Theorem 3.1 afford an intuitive interpretation: the radius scales proportionally to the confidence $g(\varphi(x))$ and inversely with the input sensitivity $\|\nabla g(\varphi(x))\|_{p,*}$. In practice, $\mathrm{Lip}_p(\varphi)$ can be made quite small as mentioned in Section 2, and furthermore the subgradient $\nabla g(\varphi(x))$ is easily evaluated as the Jacobian of $g$ at $\varphi(x)$ using standard automatic differentiation packages. This provides fast, deterministic class 1 certificates for any $\ell_p$-norm without modification of the feature-convex network's training procedure or architecture. We emphasize that our robustness certificates of Theorem 3.1 are independent of the architecture of $f$.

## 3.2 Representation Power Characterization

We now restrict our analysis to the class $\mathcal{F}_{\mathrm{Id}}$ of feature-convex classifiers with an identity feature map. This can be equivalently considered as the class of classifiers for which the input-to-logit map is convex. We therefore refer to models in $\mathcal{F}_{\mathrm{Id}}$ as *input-convex classifiers*. While the feature map $\varphi$ is useful in boosting the practical performance of our classifiers, the theoretical results in this section suggest that there is significant potential in using input-convex classifiers as a standalone solution.

**Classifying convexly separable sets.** We begin by introducing the notion of convexly separable sets, which are intimately related to decision regions representable by the class $\mathcal{F}_{\mathrm{Id}}$.

**Definition 3.3.** Let $X_1, X_2 \subseteq \mathbb{R}^d$. The ordered pair $(X_1, X_2)$ is said to be *convexly separable* if there exists a nonempty closed convex set $X \subseteq \mathbb{R}^d$ such that $X_2 \subseteq X$ and $X_1 \subseteq \mathbb{R}^d \setminus X$.

Notice that it may be the case that a pair $(X_1, X_2)$ is convexly separable yet the pair $(X_2, X_1)$ is not. Although low-dimensional intuition may raise concerns regarding the convex separability of binary-labeled data, we will soon see in Fact 3.8 and Theorem 3.10 that convex separability typically holds in high dimensions. We now show that convexly separable datasets possess the property that they may always be perfectly fit by input-convex classifiers.

**Proposition 3.4.** *For any nonempty closed convex set $X \subseteq \mathbb{R}^d$, there exists $f \in \mathcal{F}_{\mathrm{Id}}$ such that $X = f^{-1}(\{2\}) = \{x \in \mathbb{R}^d : f(x) = 2\}$. In particular, this shows that if $(X_1, X_2)$ is a convexly*

*separable pair of subsets of* $\mathbb{R}^d$, *then there exists* $f \in \mathcal{F}_{\mathrm{Id}}$ *such that* $f(x) = 1$ *for all* $x \in X_1$ *and* $f(x) = 2$ *for all* $x \in X_2$.

We also show that the converse of Proposition 3.4 holds: the geometry of the decision regions of classifiers in $\mathcal{F}_{\mathrm{Id}}$ consists of a convex set and its complement.

**Proposition 3.5.** *Let* $f \in \mathcal{F}_{\mathrm{Id}}$. *The decision region under* $f$ *associated to class* 2, *namely* $X := f^{-1}(\{2\}) = \{x \in \mathbb{R}^d : f(x) = 2\}$, *is a closed convex set.*

Note that this is not necessarily true for our more general feature-convex architectures with $\varphi \neq \mathrm{Id}$. We continue our theoretical analysis of input-convex classifiers by extending the universal approximation theorem for regressing upon real-valued convex functions (given in Chen et al. [16]) to the classification setting. In particular, Theorem 3.6 below shows that any input-convex classifier $f \in \mathcal{F}_{\mathrm{Id}}$ can be approximated arbitrarily well on any compact set by ReLU neural networks with nonnegative weights. Here, "arbitrarily well" means that the set of inputs where the neural network prediction differs from that of $f$ can be made to have arbitrarily small Lebesgue measure.

**Theorem 3.6.** *For any* $f \in \mathcal{F}_{\mathrm{Id}}$, *any compact convex subset* $X$ *of* $\mathbb{R}^d$, *and any* $\epsilon > 0$, *there exists* $\hat{f} \in \hat{\mathcal{F}}_{\mathrm{Id}}$ *such that* $m(\{x \in X : \hat{f}(x) \neq f(x)\}) < \epsilon$.

An extension of the proof of Theorem 3.6 combined with Proposition 3.4 yields that input-convex ReLU neural networks can perfectly fit convexly separable sampled datasets.

**Theorem 3.7.** *If* $(X_1, X_2)$ *is a convexly separable pair of finite subsets of* $\mathbb{R}^d$, *then there exists* $\hat{f} \in \hat{\mathcal{F}}_{\mathrm{Id}}$ *such that* $\hat{f}(x) = 1$ *for all* $x \in X_1$ *and* $\hat{f}(x) = 2$ *for all* $x \in X_2$.

Theorems 3.6 and 3.7, being specialized to models with ReLU activation functions, theoretically justify the particular parameterization in Definition 2.2 for learning feature-convex classifiers to fit convexly separable data.

**Empirical convex separability.** Interestingly, we find empirically that high-dimensional image training data is convexly separable. We illustrate this in Appendix D by attempting to reconstruct a CIFAR-10 cat image from a convex combination of the dogs and vice versa; the error is significantly positive for *every* sample in the training dataset, and image reconstruction is visually poor. This fact, combined with Theorem 3.7, immediately yields the following result.

**Fact 3.8.** *There exists* $\hat{f} \in \hat{\mathcal{F}}_{\mathrm{Id}}$ *such that* $\hat{f}$ *achieves perfect training accuracy for the unaugmented CIFAR-10 cats-versus-dogs dataset.*

The gap between this theoretical guarantee and our practical performance is large; without the feature map, our CIFAR-10 cats-dogs classifier achieves just 73.4% training accuracy (Table 4). While high training accuracy does not necessarily imply strong test set performance, Fact 3.8 demonstrates that the typical deep learning paradigm of overfitting to the training dataset is theoretically attainable [49]. We thus posit that there is substantial room for improvement in the design and optimization of input-convex classifiers. We leave the challenge of overfitting to the CIFAR-10 cats-dogs training data with an input-convex classifier as an open research problem for the field.

**Open Problem 3.9.** *Learn an input-convex* ReLU *neural network that achieves* 100% *training accuracy on the unaugmented CIFAR-10 cats-versus-dogs dataset.*

**Convex separability in high dimensions.** We conclude by investigating *why* the convex separability property that allows for Fact 3.8 may hold for natural image datasets. We argue that dimensionality facilitates this phenomenon by showing that data is easily separated by some $f \in \hat{\mathcal{F}}_{\mathrm{Id}}$ when $d$ is sufficiently large. In particular, although it may seem restrictive to rely on models in $\hat{\mathcal{F}}_{\mathrm{Id}}$ with convex class 2 decision regions, we show in Theorem 3.10 below that even uninformative data distributions that are seemingly difficult to classify may be fit by such models with high probability as the dimensionality of the data increases.

**Theorem 3.10.** *Consider* $M, N \in \mathbb{N}$. *Let* $X_1 = \{x^{(1)}, \ldots, x^{(M)}\} \subseteq \mathbb{R}^d$ *and* $X_2 = \{y^{(1)}, \ldots, y^{(N)}\} \subseteq \mathbb{R}^d$ *be samples with all elements* $x_k^{(i)}, y_l^{(j)}$ *drawn independently and identi-*

*cally from the uniform probability distribution on $[-1, 1]$. Then, it holds that*

$$\mathbb{P}\left((X_1, X_2) \text{ is convexly separable}\right) \geq \begin{cases} 1 - \left(1 - \frac{M!N!}{(M+N)!}\right)^d & \text{for all } d \in \mathbb{N}, \\ 1 & \text{if } d \geq M + N. \end{cases} \quad (2)$$

*In particular, $\hat{\mathcal{F}}_{\mathrm{Id}}$ contains an input-convex ReLU neural network that classifies all $x^{(i)}$ into class 1 and all $y^{(j)}$ into class 2 almost surely for sufficiently large dimensions $d$.*

Although the uniformly distributed data in Theorem 3.10 is unrealistic in practice, the result demonstrates that the class $\hat{\mathcal{F}}_{\mathrm{Id}}$ of input-convex ReLU neural networks has sufficient complexity to fit even the most unstructured data in high dimensions. Despite this ability, researchers have found that current input-convex neural networks tend to not overfit in practice, yielding small generalization gaps relative to conventional neural networks [60]. Achieving the modern deep learning paradigm of overfitting to the training dataset with input-convex networks is an exciting open challenge [49].

## 4 Experiments

This section compares our feature-convex classifiers against a variety of state-of-the-art baselines in the asymmetric setting. Before discussing the results, we briefly describe the datasets, baselines, and architectures used. For a more in-depth description and hyperparameter details, see Appendix E.

**Datasets.** We use four datasets. First, we consider distinguishing between $28 \times 28$ greyscale MNIST digits 3 and 8 [37], which are generally more visually similar and challenging to distinguish than other digit pairs. Next, we consider identifying malware from the "Allaple.A" class in the Malimg dataset of $512 \times 512$ bytewise encodings of malware [50]. Next, we consider distinguishing between shirts and T-shirts in the Fashion-MNIST dataset of $28 \times 28$ greyscale images [70], which tend to be the hardest classes to distinguish [33]. Finally, we consider the $32 \times 32$ RGB CIFAR-10 cat and dog images since they are relatively difficult to distinguish [26, 43, 30]. The latter two datasets can be considered as our more challenging settings. All pixel values are normalized into the interval $[0, 1]$.

**Baseline Methods.** We consider several state-of-the-art randomized and deterministic baselines. For all datasets, we evaluate the randomized smoothing certificates of Yang et al. [72] for the Gaussian, Laplacian, and uniform distributions trained with noise augmentation (denoted RS Gaussian, RS Laplacian, and RS Uniform, respectively), as well as the deterministic bound propagation framework $\alpha, \beta$-CROWN [66], which is scatter plotted since certification is only reported as a binary answer at a given radius. We also evaluate, when applicable, deterministic certified methods for each norm ball. These include the splitting-noise $\ell_1$-certificates from Levine and Feizi [40] (denoted Splitting), the orthogonality-based $\ell_2$-certificates from Trockman and Kolter [63] (denoted Cayley), and the $\ell_\infty$-distance-based $\ell_\infty$-certificates from Zhang et al. [77] (denoted $\ell_\infty$-Net). The last two deterministic methods are not evaluated on the large-scale Malimg dataset due to their prohibitive runtime. Furthermore, the $\ell_\infty$-Net was unable to significantly outperform a random classifier on the CIFAR-10 cats-dogs dataset, and is therefore only included in the MNIST 3-8 and Fashion-MNIST shirts experiments. Notice that the three randomized smoothing baselines have fundamentally different predictions and certificates than the deterministic methods (including ours), namely, the predictions are random and the certificates hold only with high probability.

**Feature-Convex Architecture.** Our simple experiments (MNIST 3-8 and Malimg) require no feature map to achieve high accuracy ($\varphi = \mathrm{Id}$). The Fashion-MNIST shirts dataset also benefited minimally from the feature map inclusion. For the CIFAR-10 cats-dogs task, we let our feature map be the concatenation $\varphi(x) = (x - \mu, |x - \mu|)$, as motivated by Appendix B, where $\mu$ is the channel-wise dataset mean (e.g., size 3 for an RGB image) broadcasted to the appropriate dimensions. Our MNIST 3-8 and Malimg architecture then consists of a simple two-hidden-layer input-convex multilayer perceptron with $(n_1, n_2) = (200, 50)$ hidden features, ReLU nonlinearities, and passthrough weights. For the Fashion-MNIST shirts (CIFAR-10 cats-dogs, resp.) dataset, we use a convex ConvNet architecture consisting of 3 (5, resp.) convolutional, BatchNorm, and ReLU layers. All models are trained using SGD on the standard binary cross entropy loss with Jacobian regularization, and clean accuracies are balanced as described in Section 1.1 and Appendix E.4 to ensure a fair comparison of different robustness certificates.

**Results and Discussion.** Experimental results for $\ell_1$-norm certification are reported in Figure 2, where our feature-convex classifier radii, denoted by Convex*, are similar or better than all other baselines across all datasets. Also reported is each method's clean test accuracy without any attacks, denoted by "clean." Due to space constraints, we defer the corresponding plots for $\ell_2$- and $\ell_\infty$-norm balls to Appendix F, where our certified radii are not dominant but still comparable to methods tailored specifically for a particular norm. We accomplish this while maintaining completely deterministic, closed-form certificates with orders-of-magnitude faster computation time than competitive baselines.

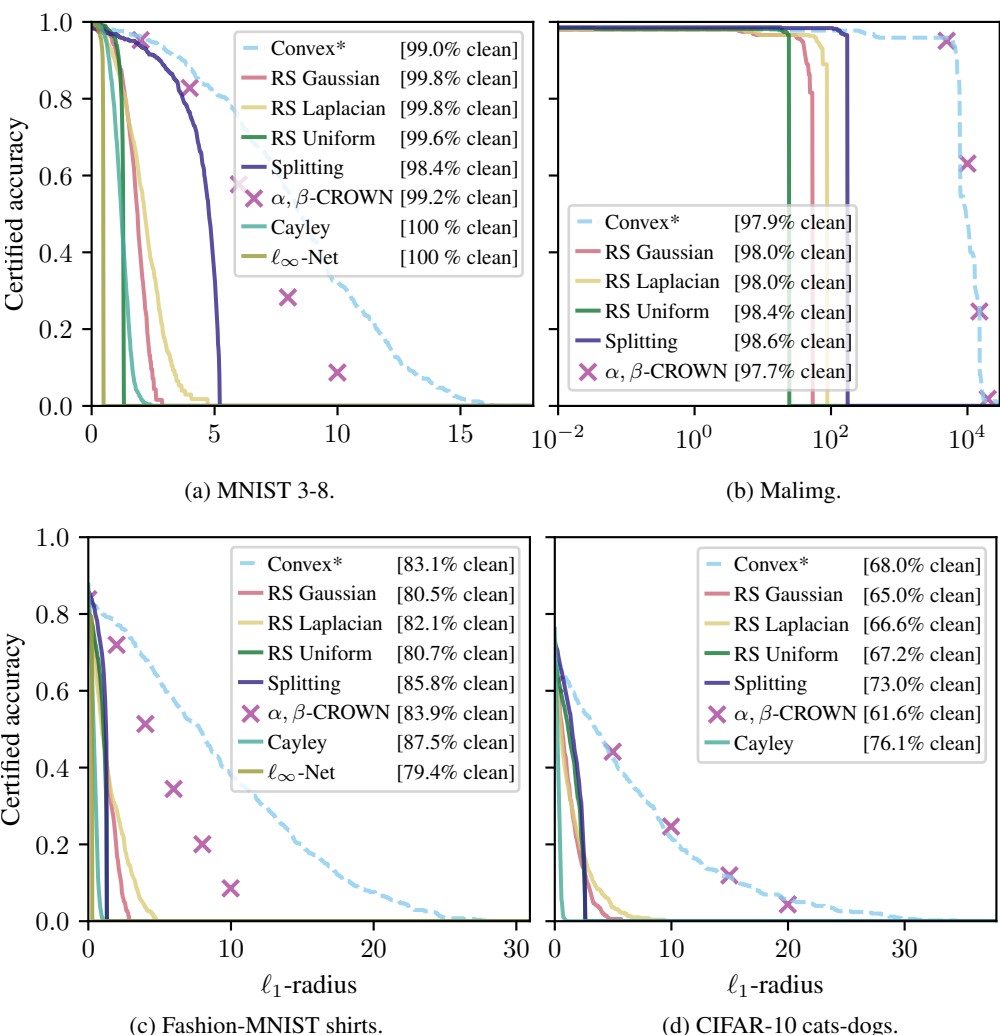

Figure 2: Class 1 certified radii curves for the $\ell_1$-norm. Note the $\log$-scale on the Malimg plot.

For the MNIST 3-8 and Malimg datasets (Figures 2a and 2b), all methods achieve high clean test accuracy. Our $\ell_1$-radii scale exceptionally well with the dimensionality of the input, with two orders of magnitude improvement over smoothing baselines for the Malimg dataset. The Malimg certificates in particular have an interesting concrete interpretation. As each pixel corresponds to one byte in the original malware file, an $\ell_1$-certificate of radius $r$ provides a robustness certificate for up to $r$ bytes in the file. Namely, even if a malware designer were to arbitrarily change $r$ malware bytes, they would be unable to fool our classifier into returning a false negative. We note that this is primarily illustrative and is unlikely to have an immediate practical impact as small semantic changes (e.g., reordering unrelated instructions) can induce large $\ell_p$-norm shifts.

While our method produces competitive robustness certificates for $\ell_2$- and $\ell_\infty$-norms (Appendix F), it offers the largest improvement for $\ell_1$-certificates in the high-dimensional image spaces considered. This is likely due to the characteristics of the subgradient dual norm factor in the denominator of

Table 1: Average runtimes (seconds) per input for computing the $\ell_1, \ell_2,$ and $\ell_\infty$-robust radii. * = our method. † = per-property verification time. ‡ = certified radius computed via binary search.

|  | MNIST 3-8 | Malimg | Fashion-MNIST shirts | CIFAR-10 cats-dogs |
|---|---|---|---|---|
| Convex* | 0.00159 | 0.00295 | 0.00180 | 0.00180 |
| RS Gaussian | 2.16 | 111.9 | 2.41 | 5.78 |
| RS Laplacian | 2.23 | 114.8 | 2.51 | 5.81 |
| RS Uniform | 2.18 | 112.4 | 2.44 | 5.80 |
| Splitting | 0.597 | 994.5 | 0.185 | 0.774 |
| $\alpha, \beta$-CROWN† | 6.088 | 6.138 | 6.425 | 9.133 |
| Cayley | 0.000505 | — | 0.0451 | 0.0441 |
| $\ell_\infty$-Net‡ | 0.138 | — | 0.115 | — |

Theorem 3.1. The dual of the $\ell_1$-norm is the $\ell_\infty$-norm, which selects the largest magnitude element in the gradient of the output logit with respect to the input pixels. As the input image scales, it is natural for the classifier to become less dependent on any one specific pixel, shrinking the denominator in Theorem 3.1. Conversely, when certifying for the $\ell_\infty$-norm, one must evaluate the $\ell_1$-norm of the gradient, which scales proportionally to the input size. Nevertheless, we find in Appendix F that our $\ell_2$- and $\ell_\infty$-radii are generally comparable those of the baselines while maintaining speed and determinism.

Our feature-convex neural network certificates are almost immediate, requiring just one forward pass and one backward pass through the network. This certification procedure requires a few milliseconds per sample on our hardware and scales well with network size. This is substantially faster than the runtime for randomized smoothing, which scales from several seconds per CIFAR-10 image to minutes for an ImageNet image [18]. The only method that rivaled our $\ell_1$-norm certificates was $\alpha, \beta$-CROWN; however, such bound propagation frameworks suffer from exponential computational complexity in network size, and even for small CIFAR-10 ConvNets typically take on the order of minutes to certify nontrivial radii. For computational tractability, we therefore used a smaller network in our experiments (Appendix E). Certification time for all methods is reported in Table 1.

Unlike the randomized smoothing baselines, our method is completely deterministic in both prediction and certification. Randomized prediction poses a particular problem for randomized smoothing certificates: even for a perturbation of a "certified" magnitude, repeated evaluations at the perturbed point will eventually yield misclassification for any nontrivial classifier. While the splitting-based certificates of Levine and Feizi [40] are deterministic, they only certify quantized (not continuous) $\ell_1$-perturbations, which scale poorly to $\ell_2$- and $\ell_\infty$-certificates (Appendix F). Furthermore, the certification runtime grows linearly in the smoothing noise $\sigma$; evaluating the certified radii at $\sigma$ used for the Malimg experiment takes several minutes per sample.

Ablation tests examining the impact of Jacobian regularization, the feature map $\varphi$, and data augmentation are included in Appendix G. We illustrate the certification performance of our method across all combinations of MNIST classes in Appendix H.

## 5 Conclusion

This work introduces the problem of asymmetric certified robustness, which we show naturally applies to a number of practical adversarial settings. We define feature-convex classifiers in this context and theoretically characterize their representation power from geometric, approximation theoretic, and statistical lenses. Closed-form sensitive-class certified robust radii for the feature-convex architecture are provided for arbitrary $\ell_p$-norms. We find that our $\ell_1$-robustness certificates in particular match or outperform those of the current state-of-the-art methods, with our $\ell_2$- and $\ell_\infty$-radii also competitive to methods tailored for a particular norm. Unlike smoothing and bound propagation baselines, we accomplish this with a completely deterministic and near-immediate computation scheme. We also show theoretically that significant performance improvements should be realizable for natural image datasets such as CIFAR-10 cats-versus-dogs. Possible directions for future research include bridging the gap between the theoretical power of feature-convex models and their practical implementation, as well as exploring more sophisticated choices of the feature map $\varphi$.

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

# Supplementary Material

## A  Classification Framework Generalization

While outside the scope of our work, we note that there are two natural ways to extend our approach to a multiclass setting with one sensitive class. Let $\mathcal{Y} = \{1, 2, \ldots, c\}$, with class 1 being the sensitive class for which we aim to generate certificates.

One approach involves a two-step architecture, where a feature-convex classifier first distinguishes between the sensitive class 1 and all other classes $\{2, 3, \ldots, c\}$ and an arbitrary second classifier distinguishes between the classes $\{2, 3, \ldots, c\}$. The first classifier could then be used to generate class 1 certificates, as described in Section 3.1.

Alternatively, we could define $g$ to map directly to $c$ output logits, with the first logit convex in the input and the other logits concave in the input. Concavity can be easily achieved by negating the output of a convex network. Let the $i$th output logit then be denoted as $g_i$ and consider an input $x$ where the classifier predicts class 1 (i.e., $g_1(\varphi(x)) \geq g_i(\varphi(x))$ for all $i \in \{2, 3, \ldots, c\}$); since the difference of a convex and a concave function is convex, we can generate a certificate for the nonnegativity of each convex decision function $g_1 \circ \varphi - g_i \circ \varphi$ around $x$. Minimizing these certificates over all $i \in \{2, 3, \ldots, c\}$ yields a robustness certificate for the sensitive class.

Note that $g$ mapping to 2 or more logits, all convex in the input, would not yield any tractable certificates. This is because the classifier decision function would now be the difference of two convex functions and have neither convex nor concave structure. We therefore choose to instantiate our binary classification networks with a single convex output logit for clarity.

### A.1  Malimg Multiclass Extension

As a proof-of-concept, we provide a concrete realization of the first scheme above on the Malimg dataset. Namely, consider the setting where we want to distinguish between "clean" binaries and 24 classes of malware. A malware designer seeks to maliciously perturb the bytes in their binary to fool a classifier into falsely predicting that the malware is "clean." We therefore consider a cascading architecture where first a feature-convex classifier answers the "clean or malware" question, and then a subsequent classifier (not necessarily feature-convex) predicts the particular class of malware in the case that the feature-convex classifier assigns a "malware" prediction. Note that, in the initial step, we can either certify the "clean" binaries or the collection of all 24 malware classes, simply by negating the feature-convex classifier output logit. We logically choose to certify the malware classes as done in our experiments of Section 4; these certificates provide guarantees against a piece of malware going undetected.

We use the same feature-convex architecture and training details as described in Appendix E. For the cascaded malware classifier, we use a ResNet-18 architecture trained with Adam for 150 epochs with a learning rate of $10^{-3}$. The confusion plot for the multiclass classifier is provided in Figure 3, with an overall accuracy of $96.5\%$. With the exception of few challenging classes to distinguish, the classifier achieves reasonable performance despite the unbalanced class sizes.

Figure 4 visualizes the distribution of certified radii for the four most common malware classes in the dataset, excluding the "Yuner.A" class which featured duplicated images. Note that certification performance varies between classes, with high correlation across different norms for a particular malware class. Classes which tend to have larger certificates can be interpreted as clustering further away from the clean binaries, requiring larger perturbations to fool the classifier.

## B  Feature Map Motivation

This section examines the importance of the feature map $\varphi$ with a low-dimensional example. Consider the binary classification setting where one class $X_2 \subseteq \mathbb{R}^d$ is clustered around the origin and the other class $X_1 \subseteq \mathbb{R}^d$ surrounds it in a ring. Here, the pair $(X_1, X_2)$ is convexly separable (see Definition 3.3) as an $\ell_2$-norm ball decision region covering $X_2$ is convex (Figure 5a). Note that the reverse pair $(X_2, X_1)$ is *not* convexly separable, as there does not exist a convex set containing $X_1$ but excluding $X_2$. A standard input-convex classifier with $\varphi = \mathrm{Id}$ would therefore be unable to

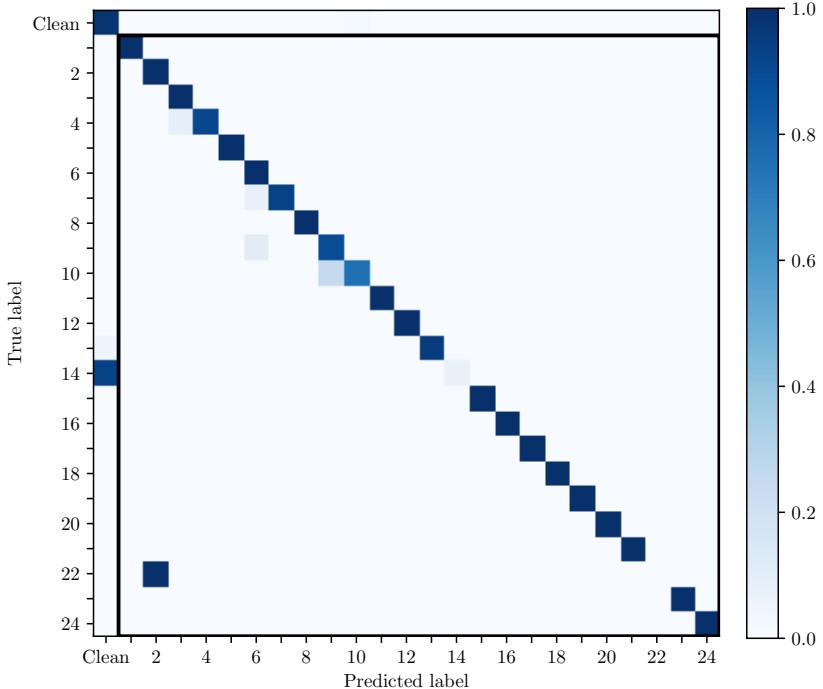

Figure 3: The row-normalized confusion plot for the Malimg multiclass classifier. The overall accuracy of the composite classifier is 96.5%. The various malware classes (1-24) are circumscribed with a black rectangle. These are certified against the class of "clean" binaries. See Section 4 for more details on the mock clean binaries.

discriminate between the classes in this direction (Proposition 3.5), i.e., we would be able to learn a classifier that generates certificates for points in $X_1$, but not $X_2$.

The above problem is addressed by choosing the feature map to be the simple concatenation $\varphi(x) = (x, |x|)$ mapping from $\mathbb{R}^d$ to $\mathbb{R}^q = \mathbb{R}^{2d}$, with associated Lipschitz constants $\mathrm{Lip}_1(\varphi) \le 2$, $\mathrm{Lip}_2(\varphi) \le \sqrt{2}$, and $\mathrm{Lip}_\infty(\varphi) \le 1$. In this augmented feature space, $X_1$ and $X_2$ are convexly separable in both directions, as they are each contained in a convex set (specifically, a half-space) whose complement contains the other class. We are now able to learn a classifier that takes $X_2$ as the sensitive class for which certificates are required (Figure 5b). This parallels the motivation of the support vector machine "kernel trick," where inputs are augmented to a higher-dimensional space wherein the data is linearly separable (instead of convexly separable as in our case).

## C   Proofs and Supplemental Theoretical Results

**Theorem 3.1.** *Let $f \in \mathcal{F}$ be as in Definition 2.1 and let $x \in f^{-1}(\{1\}) = \{x' \in \mathbb{R}^d : f(x') = 1\}$. If $\nabla g(\varphi(x)) \in \mathbb{R}^q$ is a nonzero subgradient of the convex function $g$ at $\varphi(x)$, then $f(x + \delta) = 1$ for all $\delta \in \mathbb{R}^d$ such that*

$$\|\delta\|_p < r(x) := \frac{g(\varphi(x))}{\mathrm{Lip}_p(\varphi)\|\nabla g(\varphi(x))\|_{p,*}}.$$

*Proof.* Suppose that $\nabla g(\varphi(x)) \in \mathbb{R}^q$ is a nonzero subgradient of $g$ at $\varphi(x)$, so that $g(y) \ge g(\varphi(x)) + \nabla g(\varphi(x))^\top (y - \varphi(x))$ for all $y \in \mathbb{R}^q$. Let $\delta \in \mathbb{R}^d$ be such that $\|\delta\|_p < r(x)$. Then it holds that

$$\begin{aligned}
g(\varphi(x + \delta)) &\ge g(\varphi(x)) + \nabla g(\varphi(x))^\top (\varphi(x + \delta) - \varphi(x)) \\
&\ge g(\varphi(x)) - \|\nabla g(\varphi(x))\|_{p,*}\|\varphi(x + \delta) - \varphi(x)\|_p \\
&\ge g(\varphi(x)) - \|\nabla g(\varphi(x))\|_{p,*}\,\mathrm{Lip}_p(\varphi)\|\delta\|_p \\
&> 0,
\end{aligned}$$

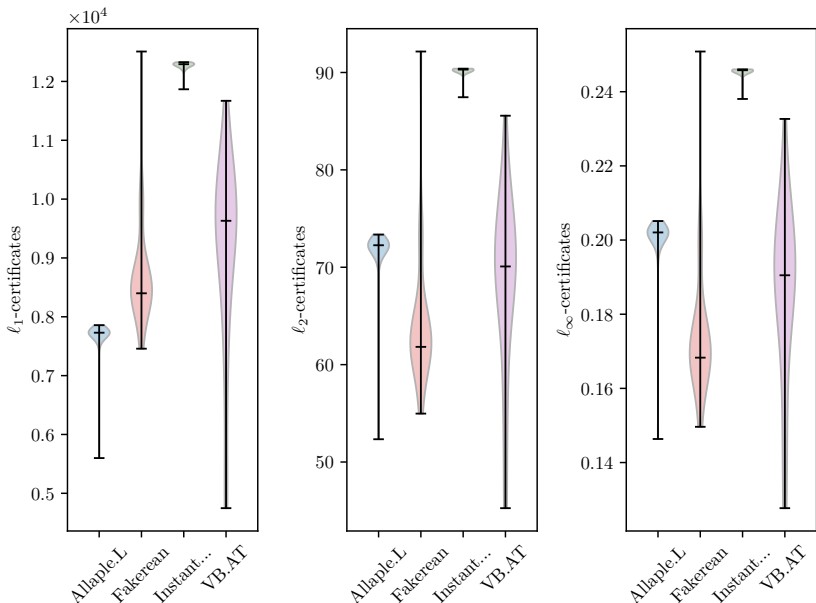

Figure 4: Certified radii distributions for four malware classes in the Malimg dataset.

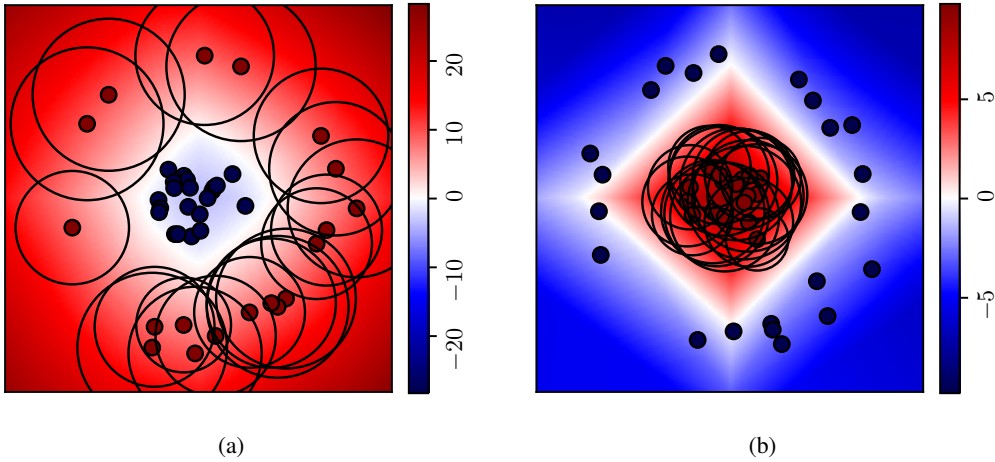

(a)                                                  (b)

Figure 5: Experiments demonstrating the role of the feature map $\varphi = (x, |x|)$ in $\mathbb{R}^2$, with the output logit shaded. Certified radii from our method are shown as black rings. (a) Certifying the outer class (dark red points). This is possible using an input-convex classifier as a convex sublevel set contains the inner class (dark blue points). (b) Certifying the inner class (dark red points). This would not be possible with $\varphi = \mathrm{Id}$ as there is no convex set containing the outer class (dark blue points) but excluding the inner. The feature map $\varphi$ enables this by permitting convex separability in the higher dimensional space. Note that although the shaded output logit is not convex in the input, we still generate certificates.

so indeed $f(x + \delta) = 1$.                                                                                    $\square$

We now introduce a preliminary lemma for the results in Section 3.2.

**Lemma C.1.** *For any nonempty closed convex set $X \subseteq \mathbb{R}^d$, there exists a convex function $g \colon \mathbb{R}^d \to \mathbb{R}$ such that $X = g^{-1}((-\infty, 0]) = \{x \in \mathbb{R}^d : g(x) \leq 0\}$.*

*Proof.* Let $X \subseteq \mathbb{R}^d$ be a nonempty closed convex set. We take the distance function $g = d_X$ defined by $d_X(x) = \inf_{y \in X} \|y - x\|_2$. Since $X$ is closed and $y \mapsto \|y - x\|_2$ is coercive for all $x \in \mathbb{R}^d$, it holds that $y \mapsto \|y - x\|_2$ attains its infimum over $X$ [10, Proposition A.8]. Let $x^{(1)}, x^{(2)} \in \mathbb{R}^d$ and let $\theta \in [0, 1]$. Then there exist $y^{(1)}, y^{(2)} \in X$ such that $g(x^{(1)}) = \|y^{(1)} - x^{(1)}\|_2$ and $g(x^{(2)}) = \|y^{(2)} - x^{(2)}\|_2$. Since $X$ is convex, it holds that $\theta y^{(1)} + (1-\theta)y^{(2)} \in X$, and therefore

$$
\begin{aligned}
g(\theta x^{(1)} + (1-\theta)x^{(2)}) &= \inf_{y \in X} \|y - (\theta x^{(1)} + (1-\theta)x^{(2)})\|_2 \\
&\leq \|\theta y^{(1)} + (1-\theta)y^{(2)} - (\theta x^{(1)} + (1-\theta)x^{(2)})\|_2 \\
&\leq \theta\|y^{(1)} - x^{(1)}\|_2 + (1-\theta)\|y^{(2)} - x^{(2)}\|_2 \\
&= \theta g(x^{(1)}) + (1-\theta)g(x^{(2)}).
\end{aligned}
$$

Hence, $g = d_X$ is convex. Since $X = \{x \in \mathbb{R}^d : \inf_{y \in X} \|y - x\|_2 = 0\} = \{x \in \mathbb{R}^d : d_X(x) = 0\} = \{x \in \mathbb{R}^d : d_X(x) \leq 0\} = \{x \in \mathbb{R}^d : g(x) \leq 0\}$ by nonnegativity of $d_X$, the lemma holds. $\square$

**Proposition 3.4.** *For any nonempty closed convex set $X \subseteq \mathbb{R}^d$, there exists $f \in \mathcal{F}_{\mathrm{Id}}$ such that $X = f^{-1}(\{2\}) = \{x \in \mathbb{R}^d : f(x) = 2\}$. In particular, this shows that if $(X_1, X_2)$ is a convexly separable pair of subsets of $\mathbb{R}^d$, then there exists $f \in \mathcal{F}_{\mathrm{Id}}$ such that $f(x) = 1$ for all $x \in X_1$ and $f(x) = 2$ for all $x \in X_2$.*

*Proof.* Let $X \subseteq \mathbb{R}^d$ be a nonempty closed convex set. By Lemma C.1, there exists a convex function $g \colon \mathbb{R}^d \to \mathbb{R}$ such that $X = \{x \in \mathbb{R}^d : g(x) \leq 0\}$. Define $f \colon \mathbb{R}^d \to \{1, 2\}$ by $f(x) = 1$ if $g(x) > 0$ and $f(x) = 2$ if $g(x) \leq 0$. Clearly, it holds that $f \in \mathcal{F}_{\mathrm{Id}}$. Furthermore, for all $x \in X$ it holds that $g(x) \leq 0$, implying that $f(x) = 2$ for all $x \in X$. Conversely, if $x \in \mathbb{R}^d$ is such that $f(x) = 2$, then $g(x) \leq 0$, implying that $x \in X$. Hence, $X = \{x \in \mathbb{R}^d : f(x) = 2\}$.

If $(X_1, X_2)$ is a convexly separable pair of subsets of $\mathbb{R}^d$, then there exists a nonempty closed convex set $X \subseteq \mathbb{R}^d$ such that $X_2 \subseteq X$ and $X_1 \subseteq \mathbb{R}^d \setminus X$, and therefore there exists $f \in \mathcal{F}_{\mathrm{Id}}$ such that $X_2 \subseteq X = f^{-1}(\{2\})$ and $X_1 \subseteq \mathbb{R}^d \setminus X = f^{-1}(\{1\})$, implying that indeed $f(x) = 1$ for all $x \in X_1$ and $f(x) = 2$ for all $x \in X_2$. $\square$

**Proposition 3.5.** *Let $f \in \mathcal{F}_{\mathrm{Id}}$. The decision region under $f$ associated to class $2$, namely $X := f^{-1}(\{2\}) = \{x \in \mathbb{R}^d : f(x) = 2\}$, is a closed convex set.*

*Proof.* For all $x \in \mathbb{R}^d$, it holds that $f(x) = 2$ if and only if $g(x) \leq 0$. Since $f \in \mathcal{F}_{\mathrm{Id}}$, $g$ is convex, and hence, $X = \{x \in \mathbb{R}^d : g(x) \leq 0\}$ is a (nonstrict) sublevel set of a convex function and is therefore a closed convex set. $\square$

In order to apply the universal approximation results in Chen et al. [16], we now introduce their parameterization of input-convex ReLU neural networks. Note that it imposes the additional constraint that the first weight matrix $A^{(1)}$ is elementwise nonnegative.

**Definition C.2.** Define $\tilde{\mathcal{F}}_{\mathrm{Id}}$ to be the class of functions $\tilde{f} \colon \mathbb{R}^d \to \{1, 2\}$ given by $\tilde{f}(x) = T(\tilde{g}(x))$ with $\tilde{g} \colon \mathbb{R}^d \to \mathbb{R}$ given by

$$
\begin{aligned}
x^{(1)} &= \mathrm{ReLU}\left(A^{(1)}x + b^{(1)}\right), \\
x^{(l)} &= \mathrm{ReLU}\left(A^{(l)}x^{(l-1)} + b^{(l)} + C^{(l)}x\right), \; l \in \{2, 3, \ldots, L-1\}, \\
\tilde{g}(x) &= A^{(L)}x^{(L-1)} + b^{(L)} + C^{(L)}x,
\end{aligned}
$$

for some $L \in \mathbb{N}$, $L > 1$, and some consistently sized matrices $A^{(1)}, C^{(1)}, \ldots, A^{(L)}, C^{(L)}$, all of which have nonnegative elements, and some consistently sized vectors $b^{(1)}, \ldots, b^{(L)}$.

The following preliminary lemma relates the class $\hat{\mathcal{F}}_{\mathrm{Id}}$ from Definition 2.2 to the class $\tilde{\mathcal{F}}_{\mathrm{Id}}$ above.

**Lemma C.3.** *It holds that $\tilde{\mathcal{F}}_{\mathrm{Id}} \subseteq \hat{\mathcal{F}}_{\mathrm{Id}}$.*

*Proof.* Let $\tilde{f} \in \tilde{\mathcal{F}}_{\mathrm{Id}}$. Then certainly $A^{(l)} \geq 0$ for all $l \in \{2, 3, \dots, L\}$, so indeed $\tilde{f} \in \hat{\mathcal{F}}_{\mathrm{Id}}$. Hence, $\tilde{\mathcal{F}}_{\mathrm{Id}} \subseteq \hat{\mathcal{F}}_{\mathrm{Id}}$. $\qquad\square$

Theorem 1 in Chen et al. [16] shows that a Lipschitz convex function can be approximated within an arbitrary tolerance. We now provide a technical lemma adapting Theorem 1 in Chen et al. [16] to show that convex functions can be *underapproximated* within an arbitrary tolerance on a compact convex subset.

**Lemma C.4.** *For any convex function* $g \colon \mathbb{R}^d \to \mathbb{R}$*, any compact convex subset* $X$ *of* $\mathbb{R}^d$*, and any* $\epsilon > 0$*, there exists* $\hat{f} \in \hat{\mathcal{F}}_{\mathrm{Id}}$ *such that* $\hat{g}(x) < g(x)$ *for all* $x \in X$ *and* $\sup_{x \in X} (g(x) - \hat{g}(x)) < \epsilon$.

*Proof.* Let $g \colon \mathbb{R}^d \to \mathbb{R}$ be a convex function, let $X$ be a compact convex subset of $\mathbb{R}^d$, and let $\epsilon > 0$. Since $g - \epsilon/2$ is a real-valued convex function on $\mathbb{R}^d$ (and hence is proper), its restriction to the closed and bounded set $X$ is Lipschitz continuous [55, Theorem 10.4], and therefore Lemma C.3 together with Theorem 1 in Chen et al. [16] gives that there exists $\hat{f} \in \tilde{\mathcal{F}}_{\mathrm{Id}} \subseteq \hat{\mathcal{F}}_{\mathrm{Id}}$ such that $\sup_{x \in X} |(g(x) - \epsilon/2) - \hat{g}(x)| < \epsilon/2$. Thus, for all $x \in X$,

$$
\begin{aligned}
g(x) - \hat{g}(x) &= \left( g(x) - \frac{\epsilon}{2} \right) - \hat{g}(x) + \frac{\epsilon}{2} \\
&> \left( g(x) - \frac{\epsilon}{2} \right) - \hat{g}(x) + \sup_{y \in X} \left| \left( g(y) - \frac{\epsilon}{2} \right) - \hat{g}(y) \right| \\
&\geq \left( g(x) - \frac{\epsilon}{2} \right) - \hat{g}(x) + \left| \left( g(x) - \frac{\epsilon}{2} \right) - \hat{g}(x) \right| \\
&\geq 0.
\end{aligned}
$$

Furthermore,

$$
\begin{aligned}
\sup_{x \in X} (g(x) - \hat{g}(x)) &= \sup_{x \in X} |g(x) - \hat{g}(x)| \\
&= \sup_{x \in X} \left| \left( g(x) - \frac{\epsilon}{2} \right) - \hat{g}(x) + \frac{\epsilon}{2} \right| \\
&\leq \sup_{x \in X} \left| \left( g(x) - \frac{\epsilon}{2} \right) - \hat{g}(x) \right| + \frac{\epsilon}{2} \\
&< \epsilon,
\end{aligned}
$$

which proves the lemma. $\qquad\square$

We leverage Lemma C.4 to construct a uniformly converging sequence of underapproximating functions.

**Lemma C.5.** *For all* $f \in \mathcal{F}_{\mathrm{Id}}$ *and all compact convex subsets* $X$ *of* $\mathbb{R}^d$*, there exists a sequence* $\{\hat{f}_n \in \hat{\mathcal{F}}_{\mathrm{Id}} : n \in \mathbb{N}\} \subseteq \hat{\mathcal{F}}_{\mathrm{Id}}$ *such that* $\hat{g}_n(x) < \hat{g}_{n+1}(x) < g(x)$ *for all* $x \in X$ *and all* $n \in \mathbb{N}$ *and* $\hat{g}_n$ *converges uniformly to* $g$ *on* $X$ *as* $n \to \infty$.

*Proof.* Let $f \in \mathcal{F}_{\mathrm{Id}}$ and let $X$ be a compact convex subset of $\mathbb{R}^d$. Let $\{\epsilon_n > 0 : n \in \mathbb{N}\}$ be a sequence such that $\epsilon_{n+1} < \epsilon_n$ for all $n \in \mathbb{N}$ and $\epsilon_n \to 0$ as $n \to \infty$. Such a sequence clearly exists, e.g., by taking $\epsilon_n = 1/n$ for all $n \in \mathbb{N}$. Now, for all $n \in \mathbb{N}$, the function $g - \epsilon_{n+1}$ is convex, and therefore by Lemma C.4 there exists $\hat{f}_n \in \hat{\mathcal{F}}_{\mathrm{Id}}$ such that $\hat{g}_n(x) < g(x) - \epsilon_{n+1}$ for all $x \in X$ and $\sup_{x \in X} ((g(x) - \epsilon_{n+1}) - \hat{g}_n(x)) < \epsilon_n - \epsilon_{n+1}$. Fixing such $\hat{f}_n, \hat{g}_n$ for all $n \in \mathbb{N}$, we see that $\sup_{x \in X} ((g(x) - \epsilon_{n+2}) - \hat{g}_{n+1}(x)) < \epsilon_{n+1} - \epsilon_{n+2}$, which implies that

$$
\hat{g}_{n+1}(x) > g(x) - \epsilon_{n+1} > \hat{g}_n(x)
$$

for all $x \in X$, which proves the first inequality. The second inequality comes from the fact that $\hat{g}_{n+1}(x) < g(x) - \epsilon_{n+2} < g(x)$ for all $x \in X$. Finally, since $g(x) - \hat{g}_n(x) > \epsilon_{n+1} > 0$ for all $x \in X$ and all $n \in \mathbb{N}$, we see that

$$
\sup_{x \in X} |g(x) - \hat{g}_n(x)| = \sup_{x \in X} (g(x) - \hat{g}_n(x)) < \epsilon_n \to 0 \text{ as } n \to \infty,
$$

which proves that $\lim_{n \to \infty} \sup_{x \in X} |g(x) - \hat{g}_n(x)| = 0$, so indeed $\hat{g}_n$ converges uniformly to $g$ on $X$ as $n \to \infty$. $\qquad\square$

With all the necessary lemmas in place, we now present our main theoretical results.

**Theorem 3.6.** *For any $f \in \mathcal{F}_{\mathrm{Id}}$, any compact convex subset $X$ of $\mathbb{R}^d$, and any $\epsilon > 0$, there exists $\hat{f} \in \hat{\mathcal{F}}_{\mathrm{Id}}$ such that $m(\{x \in X : \hat{f}(x) \neq f(x)\}) < \epsilon$.*

*Proof.* Let $f \in \mathcal{F}_{\mathrm{Id}}$ and let $X$ be a compact convex subset of $\mathbb{R}^d$. By Lemma C.5, there exists a sequence $\{\hat{f}_n \in \hat{\mathcal{F}}_{\mathrm{Id}} : n \in \mathbb{N}\} \subseteq \hat{\mathcal{F}}_{\mathrm{Id}}$ such that $\hat{g}_n(x) < \hat{g}_{n+1}(x) < g(x)$ for all $x \in X$ and all $n \in \mathbb{N}$ and $\hat{g}_n$ converges uniformly to $g$ on $X$ as $n \to \infty$. Fix this sequence.

For all $n \in \mathbb{N}$, define

$$E_n = \{x \in X : \hat{f}_n(x) \neq f(x)\},$$

i.e., the set of points in $X$ for which the classification under $\hat{f}_n$ does not agree with that under $f$. Since $\hat{g}_n(x) < g(x)$ for all $x \in X$ and all $n \in \mathbb{N}$, we see that

$$E_n = \{x \in X : \hat{g}_n(x) > 0 \text{ and } g(x) \leq 0\} \cup \{x \in X : \hat{g}_n(x) \leq 0 \text{ and } g(x) > 0\}$$
$$= \{x \in X : \hat{g}_n(x) \leq 0 \text{ and } g(x) > 0\}.$$

Since $g$ is a real-valued convex function on $\mathbb{R}^d$, it is continuous [55, Corollary 10.1.1], and therefore $g^{-1}((0, \infty)) = \{x \in \mathbb{R}^d : g(x) > 0\}$ is measurable. Similarly, $\hat{g}_n^{-1}((-\infty, 0]) = \{x \in \mathbb{R}^d : \hat{g}_n(x) \leq 0\}$ is also measurable for all $n \in \mathbb{N}$ since $\hat{g}_n$ is continuous. Furthermore, $X$ is measurable as it is compact. Therefore, $E_n$ is measurable for all $n \in \mathbb{N}$. Now, since $\hat{g}_n(x) < \hat{g}_{n+1}(x)$ for all $x \in X$ and all $n \in \mathbb{N}$, it holds that $E_{n+1} \subseteq E_n$ for all $n \in \mathbb{N}$. It is clear that to prove the result, it suffices to show that $\lim_{n \to \infty} m(E_n) = 0$. Therefore, if we show that $m\left(\bigcap_{n \in \mathbb{N}} E_n\right) = 0$, then the fact that $m(E_1) \leq m(X) < \infty$ together with Lebesgue measure's continuity from above yields that $\lim_{n \to \infty} m(E_n) = 0$, thereby proving the result.

It remains to be shown that $m\left(\bigcap_{n \in \mathbb{N}} E_n\right) = 0$. To this end, suppose for the sake of contradiction that $\bigcap_{n \in \mathbb{N}} E_n \neq \emptyset$. Then there exists $x \in \bigcap_{n \in \mathbb{N}} E_n$, meaning that $g(x) > 0$ and $\hat{g}_n(x) \leq 0$ for all $n \in \mathbb{N}$. Thus, for this $x \in X$, we find that $\limsup_{n \to \infty} \hat{g}_n(x) \leq 0 < g(x)$, which contradicts the fact that $\hat{g}_n$ uniformly converges to $g$ on $X$. Therefore, it must be that $\bigcap_{n \in \mathbb{N}} E_n = \emptyset$, and thus $m\left(\bigcap_{n \in \mathbb{N}} E_n\right) = 0$, which concludes the proof. $\square$

**Theorem 3.7.** *If $(X_1, X_2)$ is a convexly separable pair of finite subsets of $\mathbb{R}^d$, then there exists $\hat{f} \in \hat{\mathcal{F}}_{\mathrm{Id}}$ such that $\hat{f}(x) = 1$ for all $x \in X_1$ and $\hat{f}(x) = 2$ for all $x \in X_2$.*

*Proof.* Throughout this proof, we denote the complement of a set $Y \subseteq \mathbb{R}^d$ by $Y^c = \mathbb{R}^d \setminus Y$.

Suppose that $X_1 = \{x^{(1)}, \ldots, x^{(M)}\} \subseteq \mathbb{R}^d$ and $X_2 = \{y^{(1)}, \ldots, y^{(N)}\} \subseteq \mathbb{R}^d$ are such that $(X_1, X_2)$ is convexly separable. Then, by definition of convex separability, there exists a nonempty closed convex set $X' \subseteq \mathbb{R}^d$ such that $X_2 \subseteq X'$ and $X_1 \subseteq \mathbb{R}^d \setminus X'$. Let $X = X' \cap \mathrm{conv}(X_2)$. Since $X_2 \subseteq X'$ and both sets $X'$ and $\mathrm{conv}(X_2)$ are convex, the set $X$ is nonempty and convex. By finiteness of $X_2$, the set $\mathrm{conv}(X_2)$ is compact, and therefore by closedness of $X'$, the set $X$ is compact and hence closed.

By Proposition 3.4, there exists $f \in \mathcal{F}_{\mathrm{Id}}$ such that $f^{-1}(\{2\}) = X$. Since $\mathrm{conv}(X_1 \cup X_2)$ is compact and convex, Lemma C.5 gives that there exists a sequence $\{\hat{f}_n \in \hat{\mathcal{F}}_{\mathrm{Id}} : n \in \mathbb{N}\} \subseteq \hat{\mathcal{F}}_{\mathrm{Id}}$ such that $\hat{g}_n(x) < \hat{g}_{n+1}(x) < g(x)$ for all $x \in \mathrm{conv}(X_1 \cup X_2)$ and all $n \in \mathbb{N}$ and $\hat{g}_n$ converges uniformly to $g$ on $\mathrm{conv}(X_1 \cup X_2)$ as $n \to \infty$. Fix this sequence.

Let $x \in X_2$. Then, since $X_2 \subseteq X'$ and $X_2 \subseteq \mathrm{conv}(X_2)$, it holds that $x \in X' \cap \mathrm{conv}(X_2) = X = f^{-1}(\{2\})$, implying that $f(x) = 2$ and hence $g(x) \leq 0$. Since $\hat{g}_n(x) < g(x)$ for all $n \in \mathbb{N}$, this shows that $\hat{f}_n(x) = 2$ for all $n \in \mathbb{N}$. On the other hand, let $i \in \{1, \ldots, M\}$ and consider $x = x^{(i)} \in X_1$. Since $X_1 \subseteq \mathbb{R}^d \setminus X' = \mathbb{R}^d \cap (X')^c \subseteq \mathbb{R}^d \cap (X' \cap \mathrm{conv}(X_2))^c = \mathbb{R}^d \cap X^c = \mathbb{R}^d \cap f^{-1}(\{1\})$, it holds that $f(x) = 1$ and thus $g(x) > 0$. Suppose for the sake of contradiction that $\hat{f}_n(x) = 2$ for all $n \in \mathbb{N}$. Then $\hat{g}_n(x) \leq 0$ for all $n \in \mathbb{N}$. Therefore, for this $x \in X_1$, we find that $\limsup_{n \to \infty} \hat{g}_n(x) \leq 0 < g(x)$, which contradicts the fact that $\hat{g}_n$ uniformly converges to $g$ on $\mathrm{conv}(X_1 \cup X_2)$. Therefore, it must be that there exists $n_i \in \mathbb{N}$ such that $\hat{f}_{n_i}(x) = 1$, and thus $\hat{g}_{n_i}(x) > 0$. Since $\hat{g}_n(x) < \hat{g}_{n+1}(x)$ for all $n \in \mathbb{N}$, this implies that $\hat{g}_n(x) > 0$ for all $n \geq n_i$. Hence, $\hat{f}_n(x) = \hat{f}_n(x^{(i)}) = 1$ for all $n \geq n_i$.

Let $n^\star$ be the maximum of all such $n_i$, i.e., $n^\star = \max\{n_i : i \in \{1, \ldots, M\}\}$. Then the above analysis shows that $\hat{f}_{n^\star}(x) = 2$ for all $x \in X_2$ and that $\hat{f}_{n^\star}(x) = 1$ for all $x \in X_1$. Since $\hat{f}_{n^\star} \in \hat{\mathcal{F}}_{\mathrm{Id}}$, the claim has been proven. $\qquad\square$

**Theorem 3.10.** *Consider $M, N \in \mathbb{N}$. Let $X_1 = \{x^{(1)}, \ldots, x^{(M)}\} \subseteq \mathbb{R}^d$ and $X_2 = \{y^{(1)}, \ldots, y^{(N)}\} \subseteq \mathbb{R}^d$ be samples with all elements $x_k^{(i)}, y_l^{(j)}$ drawn independently and identically from the uniform probability distribution on $[-1, 1]$. Then, it holds that*

$$
\mathbb{P}\left((X_1, X_2) \text{ is convexly separable}\right) \geq 
\begin{cases}
1 - \left(1 - \frac{M!N!}{(M+N)!}\right)^d & \text{for all } d \in \mathbb{N}, \\
1 & \text{if } d \geq M + N.
\end{cases}
\tag{2}
$$

*In particular, $\hat{\mathcal{F}}_{\mathrm{Id}}$ contains an input-convex* ReLU *neural network that classifies all $x^{(i)}$ into class $1$ and all $y^{(j)}$ into class $2$ almost surely for sufficiently large dimensions $d$.*

*Proof.* Throughout the proof, we denote the cardinality of a set $S$ by $|S|$. For the reader's convenience, we also recall that, for $n \in \mathbb{N}$, the symmetric group $S_n$ consists of all permutations (i.e., bijections) on the set $\{1, 2, \ldots, n\}$, and that $|S_n| = n!$. If $\sigma\colon \{1, 2, \ldots, n\} \to \{1, 2, \ldots, n\}$ is a permutation in $S_n$, we denote the restriction of $\sigma$ to the domain $I \subseteq \{1, 2, \ldots, n\}$ by $\sigma|_I\colon I \to \{1, 2, \ldots, n\}$, which we recall is defined by $\sigma|_I(i) = \sigma(i)$ for all $i \in I$, and is not necessarily a permutation on $I$ in general.

Consider first the case where $d \geq M + N$. Let $b \in \mathbb{R}^{M+N}$ be the vector defined by $b_i = 1$ for all $i \in \{1, \ldots, M\}$ and $b_i = -1$ for all $i \in \{M + 1, \ldots, M + N\}$. Then, since $x_k^{(i)}, y_l^{(j)}$ are independent uniformly distributed random variables on $[-1, 1]$, it holds that the matrix

$$
\begin{bmatrix}
x^{(1)^\top} \\
\vdots \\
x^{(M)^\top} \\
y^{(1)^\top} \\
\vdots \\
y^{(N)^\top}
\end{bmatrix}
\in \mathbb{R}^{(M+N) \times d}
$$

has rank $M + N$ almost surely, and therefore the linear system of equations

$$
\begin{bmatrix}
x^{(1)^\top} \\
\vdots \\
x^{(M)^\top} \\
y^{(1)^\top} \\
\vdots \\
y^{(N)^\top}
\end{bmatrix}
a = b
$$

has a solution $a \in \mathbb{R}^d$ with probability $1$, and we note that from this solution we find that $X_2$ is a subset of the nonempty closed convex set $\{x \in \mathbb{R}^d : a^\top x \leq 0\}$ and that $X_1$ is a subset of its complement. Hence, $(X_1, X_2)$ is convexly separable with probability $1$ in this case.

Now let us consider the general case: $d \in \mathbb{N}$ and in general it may be the case that $d < M + N$. For notational convenience, let $P$ be the probability of interest:

$$
P = \mathbb{P}\left((X_1, X_2) \text{ is convexly separable}\right).
$$

Suppose that there exists a coordinate $k \in \{1, 2, \ldots, d\}$ such that $x_k^{(i)} < y_k^{(j)}$ for all pairs $(i, j) \in \{1, 2, \ldots, M\} \times \{1, 2, \ldots, N\}$ and that $a := \min\{y_k^{(1)}, \ldots, y_k^{(N)}\} < \max\{y_k^{(1)}, \ldots, y_k^{(N)}\} =: b$. Then, let $X = \{x \in \mathbb{R}^d : x_k \in [a, b]\}$. That is, $X$ is the extrusion of the convex hull of the projections $\{y_k^{(1)}, \ldots, y_k^{(N)}\}$ along all remaining coordinates. The set $X$ is a nonempty closed convex set, and it

is clear by our supposition that $X_2 \subseteq X$ and $X_1 \subseteq \mathbb{R}^d \setminus X$. Therefore, the supposition implies that $(X_1, X_2)$ is convexly separable, and thus

$$P \geq \mathbb{P} \left( \text{there exists } k \in \{1, 2, \ldots, d\} \text{ such that } x_k^{(i)} < y_k^{(j)} \text{ for all pairs } (i, j) \right.$$
$$\left. \text{and that } \min\{y_k^{(1)}, \ldots, y_k^{(N)}\} < \max\{y_k^{(1)}, \ldots, y_k^{(N)}\} \right)$$
$$= 1 - \mathbb{P} \left( \text{for all } k \in \{1, 2, \ldots, d\}, \text{ it holds that } x_k^{(i)} \geq y_k^{(j)} \text{ for some pair } (i, j) \right.$$
$$\left. \text{or that } \min\{y_k^{(1)}, \ldots, y_k^{(N)}\} = \max\{y_k^{(1)}, \ldots, y_k^{(N)}\} \right)$$
$$= 1 - \prod_{k=1}^{d} \mathbb{P} \left( x_k^{(i)} \geq y_k^{(j)} \text{ for some pair } (i, j) \text{ or } \min\{y_k^{(1)}, \ldots, y_k^{(N)}\} = \max\{y_k^{(1)}, \ldots, y_k^{(N)}\} \right),$$

where the final equality follows from the independence of the coordinates of the samples. Since $\min\{y_k^{(1)}, \ldots, y_k^{(N)}\} < \max\{y_k^{(1)}, \ldots, y_k^{(N)}\}$ almost surely, we find that

$$P \geq 1 - \prod_{k=1}^{d} \left( \mathbb{P}(x_k^{(i)} \geq y_k^{(j)} \text{ for some pair } (i, j)) \right.$$
$$\left. + \mathbb{P}(\min\{y_k^{(1)}, \ldots, y_k^{(N)}\} = \max\{y_k^{(1)}, \ldots, y_k^{(N)}\}) \right)$$
$$= 1 - \prod_{k=1}^{d} \mathbb{P}(x_k^{(i)} \geq y_k^{(j)} \text{ for some pair } (i, j))$$
$$= 1 - \prod_{k=1}^{d} \left( 1 - \mathbb{P}(x_k^{(i)} < y_k^{(j)} \text{ for all pairs } (i, j)) \right) \tag{3}$$
$$= 1 - \prod_{k=1}^{d} \left( 1 - \mathbb{P} \left( \max_{i \in \{1, 2, \ldots, M\}} x_k^{(i)} < \min_{j \in \{1, 2, \ldots, N\}} y_k^{(j)} \right) \right)$$
$$= 1 - \prod_{k=1}^{d} \left( 1 - \mathbb{P} \left( (x_k^{(1)}, \ldots, x_k^{(M)}, y_k^{(1)}, \ldots, y_k^{(N)}) \in \bigcup_{\sigma \in S} E_\sigma \right) \right),$$

where we define $S$ to be the set of permutations on $\{1, \ldots, M + N\}$ whose restriction to $\{1, \ldots, M\}$ is also a permutation;

$$S = \left\{ \sigma \in S_{M+N} : \sigma|_{\{1, \ldots, M\}} \in S_M \right\},$$

and where, for a permutation $\sigma \in S_{M+N}$, $E_\sigma$ is the event where an $(M + N)$-vector has indices ordered according to $\sigma$;

$$E_\sigma = \{z \in \mathbb{R}^{M+N} : z_{\sigma(1)} < \cdots < z_{\sigma(M+N)}\}.$$

We note that the final equality in (3) relies on the fact that $\mathbb{P}(x_k^{(i)} = x_k^{(i')}) = \mathbb{P}(y_k^{(j)} = y_k^{(j')}) = 0$ for all $i' \neq i$ and all $j' \neq j$, which is specific to our uniform distribution at hand.

Now, since $E_\sigma, E_{\sigma'}$ are disjoint for distinct permutations $\sigma, \sigma' \in S_{M+N}$, the bound (3) gives that

$$P \geq 1 - \prod_{k=1}^{d} \left( 1 - \sum_{\sigma \in S} \mathbb{P}((x_k^{(1)}, \ldots, x_k^{(M)}, y_k^{(1)}, \ldots, y_k^{(N)}) \in E_\sigma) \right). \tag{4}$$

Since $x_k^{(1)}, \ldots, x_k^{(M)}, y_k^{(1)}, \ldots, y_k^{(N)}$ are independent and identically distributed samples, they define an exchangeable sequence of random variables, implying that $\mathbb{P}((x_k^{(1)}, \ldots, x_k^{(M)}, y_k^{(1)}, \ldots, y_k^{(N)}) \in E_\sigma) = \mathbb{P}(x_k^{(1)} < \cdots < x_k^{(M)} < y_k^{(1)} < \cdots < y_k^{(N)})$ for all permutations $\sigma \in S_{M+N}$. Since, under the uniform distribution at hand, $(x_k^{(1)}, \ldots, x_k^{(M)}, y_k^{(1)}, \ldots, y_k^{(N)}) \in E_\sigma$ for some $\sigma \in S_{M+N}$ almost

surely, it holds that

$$1 = \mathbb{P}\left((x_k^{(1)}, \ldots, x_k^{(M)}, y_k^{(N)}, \ldots, y_k^{(N)}) \in \bigcup_{\sigma \in S_{M+N}} E_\sigma\right)$$

$$= \sum_{\sigma \in S_{M+N}} \mathbb{P}((x_k^{(1)}, \ldots, x_k^{(M)}, y_k^{(1)}, \ldots, y_k^{(N)}) \in E_\sigma)$$

$$= |S_{M+N}| \, \mathbb{P}(x_k^{(1)} < \cdots x_k^{(M)} < y_k^{(1)} < \cdots < y_k^{(N)}).$$

This implies that

$$\mathbb{P}((x_k^{(1)}, \ldots, x_k^{(M)}, y_k^{(1)}, \ldots, y_k^{(N)}) \in E_\sigma) = \frac{1}{|S_{M+N}|} = \frac{1}{(M+N)!}$$

for all permutations $\sigma \in S_{M+N}$. Hence, our bound (4) becomes

$$P \geq 1 - \prod_{k=1}^{d}\left(1 - \frac{|S|}{(M+N)!}\right) = 1 - \left(1 - \frac{|S|}{(M+N)!}\right)^d.$$

Finally, we immediately see that that map $\Gamma \colon S_M \times S_N \to S_{M+N}$ defined by

$$\Gamma(\sigma, \sigma')(i) = \begin{cases} \sigma(i) & \text{if } i \in \{1, \ldots, M\}, \\ \sigma'(i - M) + M & \text{if } i \in \{M+1, \ldots, M+N\}, \end{cases}$$

is injective and has image $S$, implying that $|S| = |S_M \times S_N| = |S_M||S_N| = M!N!$. Thus,

$$P \geq 1 - \left(1 - \frac{M!N!}{(M+N)!}\right)^d,$$

which proves (2).

The unit probability of $\hat{\mathcal{F}}_{\mathrm{Id}}$ containing a classifier that classifies all $x^{(i)}$ into class 1 and all $y^{(j)}$ into class 2 for large $d$ follows immediately from Theorem 3.7. $\qquad\square$

## D   CIFAR-10 Cats-versus-Dogs Convex Separability

In order to establish that the cat and dog images in CIFAR-10 are convexly separable, we experimentally attempt to reconstruct an image from one class using a convex combination of all images in the other class (without augmentation such as random crops, flips, etc.). Namely, if $x$ is drawn from one class and $y^{(1)}, \ldots, y^{(N)}$ represent the entirety of the other class, we form the following optimization problem:

$$\begin{aligned} \underset{\alpha \in \mathbb{R}^N}{\text{minimize}} \quad & \left\| x - \sum_{j=1}^{N} \alpha_j y^{(j)} \right\|_2 \\ \text{subject to} \quad & \alpha \geq 0, \\ & \sum_{j=1}^{N} \alpha_j = 1. \end{aligned}$$

The reverse experiment for the other class follows similarly. We solve the optimization using MOSEK [6], and report the various norms of $x - \sum_{j=1}^{N} \alpha_j y^{(j)}$ in Figure 6. Reconstruction accuracy is generally very poor, with no reconstruction achieving better than an $\ell_1$-error of $52$. A typical reconstructed image is shown in Figure 7.

Yousefzadeh [74] and Balestriero et al. [9] showed a related empirical result for CIFAR-10, namely, that no test set image can be reconstructed as a convex combination of training set images. However, we remark that their findings do not necessarily imply that a training set image cannot be reconstructed via other training set images; our new finding that the CIFAR-10 cats-versus-dogs training set is convexly separable is required in order to assert Fact 3.8.

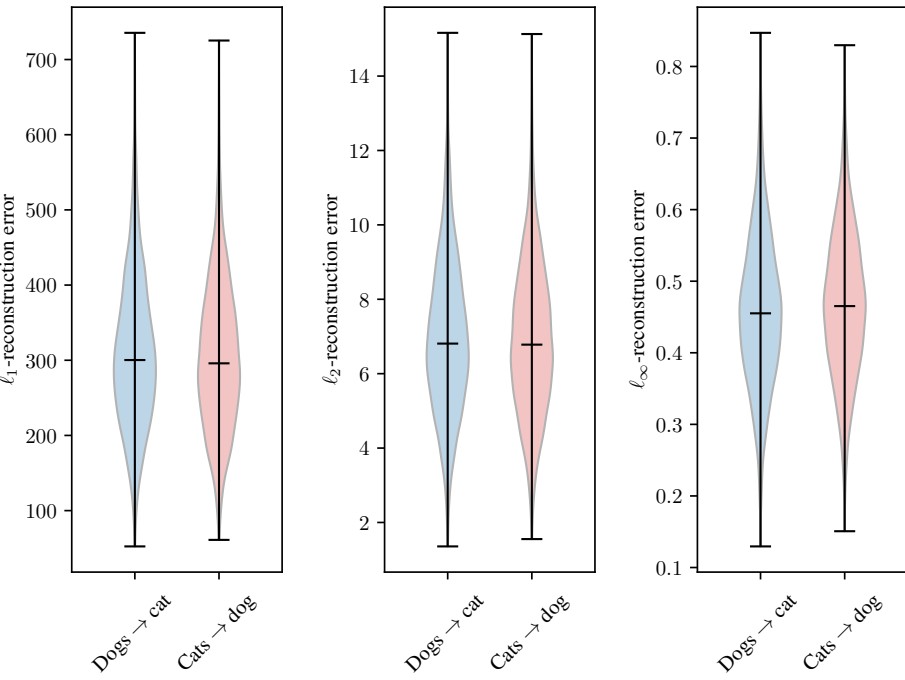

Figure 6: Reconstructing CIFAR-10 cat and dog images as convex combinations. The label "Dogs → cat" indicates that a cat image was attempted to be reconstructed as a convex combination of all 5000 dog images.

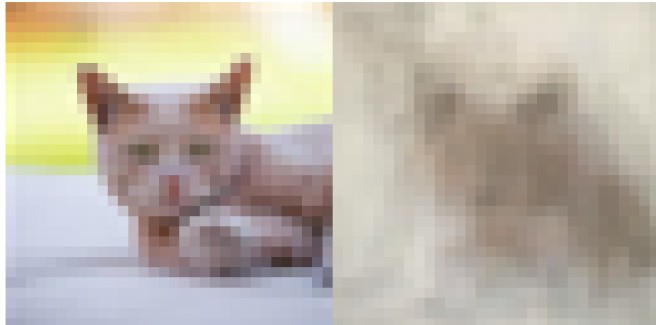

Figure 7: Reconstructing a CIFAR-10 cat image (left) from a convex combination of dog images (right). The reconstruction error norms are 294.57, 6.65, and 0.38 for the $\ell_1$-, $\ell_2$-, and $\ell_\infty$-norms, respectively. These are typical, as indicated by Figure 6.

# E   Experimental Setup

We include a detailed exposition of our experimental setup in this section, beginning with general details on our choice of epochs and batch size. We then discuss baseline methods, architecture choices for our method, class balancing, and data processing.

**Epochs and batch size.** Exempting the randomized smoothing baselines, for the MNIST 3-8 and Fashion-MNIST shirts experiments, we use 60 epochs for all methods. This is increased to 150 epochs for the Malimg dataset and CIFAR-10 cats-dogs experiments. The batch size is 64 for all datasets besides the $512 \times 512$ Malimg dataset, where it is lowered to 32.

To ensure a fair comparison, the randomized smoothing baseline epochs are scaled larger than the aforementioned methods according to the noise value specified in the sweeps in Section I. The final epochs and smoothing noise values used are reported in Table 2. Note that as classifiers are typically more robust to the noise from splitting smoothing, larger values of $\sigma$ are used for only this smoothing method in the MNIST 3-8 and Malimg datasets. For Malimg, we find experimentally that even noise values of up to $\sigma = 100$ are tractable for the splitting method, outside the sweep range considered in Section I. As verification at that $\sigma$ already takes several minutes per sample and runtime scales linearly with $\sigma$, we do not explore larger values of $\sigma$.

Table 2: Randomized smoothing final noise and epoch hyperparameters.

| Dataset | Laplacian, Uniform, Gaussian Parameters | Splitting Parameters |
|---|---|---|
| MNIST 3-8 | $(\sigma, n) = (0.75, 60)$ | $(\sigma, n) = (0.75 \cdot 4, 60 \cdot 4)$ |
| Malimg | $(\sigma, n) = (3.5 \cdot 4, 150 \cdot 4)$ | $(\sigma, n) = (100, 150 \cdot 4)$ |
| Fashion-MNIST shirts | $(\sigma, n) = (0.75, 60)$ | $(\sigma, n) = (0.75, 60)$ |
| CIFAR-10 cats-dogs | $(\sigma, n) = (0.75 \cdot 2, 600 \cdot 2)$ | $(\sigma, n) = (0.75 \cdot 2, 600 \cdot 2)$ |

**Hardware.** All experiments were conducted on a single Ubuntu 20.04 instance with an Nvidia RTX A6000 GPU. Complete reproduction of the experiments takes approximately 0.08 GPU-years.

### E.1 Datasets

We introduce the various datasets considered in this work. MNIST 3-8 and Malimg are relatively simple classification problems where near-perfect classification accuracy is attainable; the Malimg dataset falls in this category despite containing relatively large images. Our more challenging settings consist of a Fashion-MNIST shirts dataset as well as CIFAR-10 cats-versus-dogs dataset.

For consistency with [77], we augment the MNIST and Fashion-MNIST training data with 1-pixel padding and random cropping. The CIFAR-10 dataset is augmented with 3-pixel edge padding, horizontal flips, and random cropping. The Malimg dataset is augmented with 20-pixel padding and random $512 \times 512$ cropping.

For CIFAR-10, MNIST, and Fashion-MNIST, we use the preselected test sets. For Malimg we hold out a random 20% test dataset, although this may not be entirely used during testing. The training set is further subdivided by an 80%-20% validation split. For all experiments, we normalize pixel values into the interval $[0, 1]$, and we use the first 1000 test samples to evaluate our methods.

**MNIST 3-8.** For our MNIST binary classification problem, we choose the problem of distinguishing between 3 and 8 [37]. These were selected as 3 and 8 are generally more visually similar and challenging to distinguish than other digit pairs. Images are $28 \times 28$ pixels and greyscale.

**Malimg.** Our malware classification experiments use greyscale, bytewise encodings of raw malware binaries Nataraj et al. [50]. Each image pixel corresponds to one byte of data, in the range of 0–255, and successive bytes are added horizontally from left to right on the image until wrapping at some predetermined width. We use the extracted malware images from the seminal dataset Nataraj et al. [50], normalizing pixel values into $[0, 1]$, as well as padding and cropping images to be $512 \times 512$. Note that licensing concerns generally prevent the distribution of "clean" executable binaries. As this work is focused on providing a general approach to robust classification, in the spirit of reproducibility we instead report classification results between different kinds of malware. Namely, we distinguish between malware from the most numerous "Allaple.A" class (2949 samples) and an identically-sized random subset of all other 24 malware classes. To simulate a scenario where we must provide robustness against evasive malware, we provide certificates for the latter collection of classes.

**Fashion-MNIST shirts.** The hardest classes to distinguish in the Fashion-MNIST dataset are T-shirts vs shirts, which we take as our two classes [33, 70]. Images are $28 \times 28$ pixels and greyscale.

**CIFAR-10 cats-dogs.** We take as our two CIFAR-10 classes the cat and dog classes since they are relatively difficult to distinguish [26, 43, 30]. Other classes (e.g., ships) are typically easier to classify since large background features (e.g., blue water) are strongly correlated with the target label. Samples are $32 \times 32$ RGB images.

### E.2 Baseline Methods

We provide additional details on each of the baseline methods below.

**Randomized smoothing.** Since the certification runtime of randomized smoothing is large, especially for the $512 \times 512$ pixel Malimg images, we evaluate the randomized smoothing classifiers over $10^4$ samples and project the certified radius to $10^5$ samples by scaling the number fed into the Clopper-Pearson confidence interval, as described in [18]. This allows for a representative and improved certified accuracy curve while dramatically reducing the method's runtime. We take an initial guess for the certification class with $n_0 = 100$ samples and set the incorrect prediction tolerance parameter $\alpha = 0.001$. For CIFAR-10 we use a depth-40 Wide ResNet base classifier, mirroring the choices from Cohen et al. [18], Yang et al. [72]; for all other datasets we use a ResNet-18. All networks are trained using SGD with an initial learning rate of $0.1$, Nesterov momentum of $0.9$, weight decay of $10^{-4}$, and cosine annealing scheduling as described in Yang et al. [72]. Final smoothing noise values are selected as in Table 2, and are determined from the noise level comparison sweeps in Appendix I.

**Splitting noise.** As this method is a deterministic derivative of randomized smoothing, it avoids the many aforementioned hyperparameter choices. We use the same architectures described above for the other randomized smoothing experiments.

**Cayley convolutions.** To maintain consistency, we use a two-hidden-layer multilayer perceptron with $(n_1, n_2) = (200, 50)$ hidden features, CayleyLinear layers, and GroupSort activations for the MNIST experiment. For the more challenging Fashion-MNIST and CIFAR-10 experiments, we use the ResNet-9 architecture implementation from [63]. Following the authors' suggestions, we train these networks using Adam with a learning rate of $0.001$.

$\ell_\infty$**-distance nets.** As the architecture of the $\ell_\infty$-distance net [77] is substantially different from traditional architectures, we use the authors' 5-layer MNIST/Fashion-MNIST architecture and 6-layer CIFAR-10 architecture with 5120 neurons per hidden layer. Unfortunately, the classification accuracy on the CIFAR-10 cats-dogs experiment remained near $50\%$ throughout training. This was not the case when we tested easier classes, such as planes-versus-cars, where large features (e.g., blue sky) can be used to discriminate. We therefore only include this model in the MNIST and Fashion-MNIST experiments, and use the training procedure directly from the aforementioned paper's codebase.

$\alpha, \beta$**-CROWN.** As $\alpha, \beta$-CROWN certification time scales exponentially with the network size, we keep the certified networks small in order to improve the certification performance of the baseline. For all datasets, we train and certify a one-hidden-layer network with 200 hidden units and $\mathrm{ReLU}$ activations. All networks are adversarially trained for a $\ell_\infty$-perturbation radius starting at $0.001$ and linearly scaling to the desired $\epsilon$ over the first 20 epochs, as described in Kayed et al. [33], which trained the models used in Wang et al. [66]. The desired final $\epsilon$ is set to $0.3$ for MNIST, $0.1$ for Fashion-MNIST and Malimg, and $2/255$ for CIFAR-10. The adversarial training uses a standard PGD attack with 50 steps and step size $2\epsilon/50$. Other optimizer training details are identical to Wang et al. [66]. The branch-and-bound timeout is set to 30 seconds to maintain comparability to other methods, and robustness is evaluated over a dataset-dependent range of discrete radii for each adversarial norm.

### E.3 Feature-Convex Architecture and Training

In this section, we provide more details of the feature-convex architectures used in our experiments of Section 4. For the more challenging datasets (Fashion-MNIST shirts and CIFAR-10 cats-dogs), we use various instantiations of a convex ConvNet (described below) where successive layers have a constant number of channels and image size. This allows for the addition of identity residual connections to each convolution and lets us remove the passthrough connections altogether. Convexity is enforced by projecting relevant weights onto the nonnegative orthant after each epoch and similarly constraining BatchNorm $\gamma$ parameters to be positive. We initialize positive weight matrices to be drawn uniformly from the interval $[0, \epsilon]$, where $\epsilon = 0.003$ for linear weights and $\epsilon = 0.005$ for convolutional weights. Jacobian regularization is also used to improve our certified radii [31].

The convex ConvNet architecture consists of a sequence of convolutional layers, BatchNorms, and $\mathrm{ReLU}$ nonlinearities. The first convolutional layer is unconstrained, as the composition of a convex function with an affine function is still convex [2]. All subsequent convolutions and the final linear readout layer are uniformly initialized from some small positive weight interval ($[0, 0.003]$ for linear weights, $[0, 0.005]$ for convolutional weights) and projected to have nonnegative weights after

each gradient step. We found this heuristic initialization choice helps to stabilize network training, as standard Kaiming initialization assumptions are violated when weights are constrained to be nonnegative instead of normally distributed with mean zero. More principled weight initialization strategies for this architecture would form an exciting area of future research. Before any further processing, inputs into the network are fed into an initial BatchNorm—this enables flexibility with different feature augmentation maps.

Since the first convolutional layer is permitted negative weights, we generally attain better performance by enlarging the first convolution kernel size (see Table 3). For subsequent convolutions, we set the stride to 1, the input and output channel counts to the output channel count from the first convolution, and the padding to half the kernel size, rounded down. This ensures that the output of each of these deeper convolutions has equivalent dimension to its input, allowing for an identity residual connection across each convolution. If $C_i(z)$ is a convolutional operation on a hidden feature $z$, this corresponds to evaluating $C_i(z) + z$ instead of just $C_i(z)$. The final part of the classifier applies MaxPool and BatchNorm layers before a linear readout layer with output dimension 1. See Figure 8 for a diagram depicting an exemplar convex ConvNet instantiation.

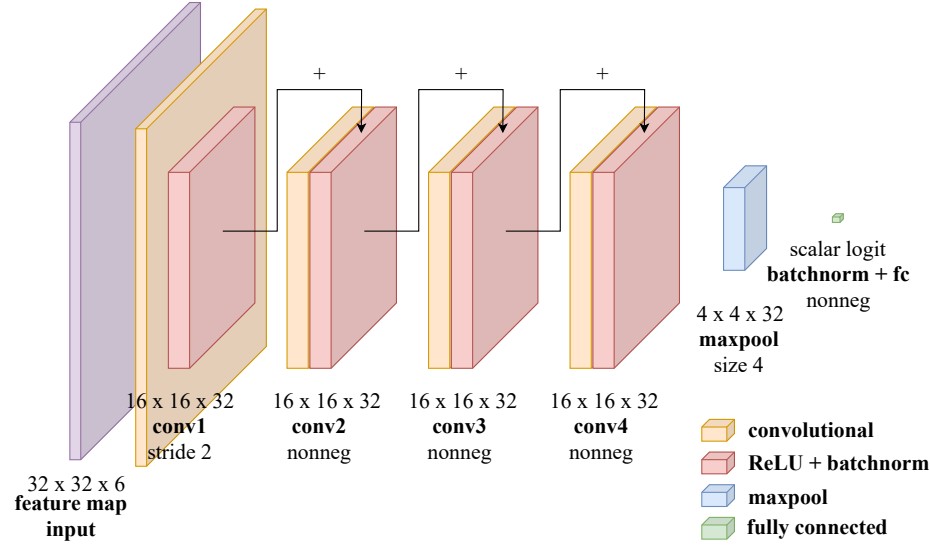

Figure 8: An example convex ConvNet of depth $4$ with a $C_1$ stride of 2, pool size of 4, and $32 \times 32$ RGB images. There are 6 input channels from the output of the feature map $\varphi \colon x \mapsto (x - \mu, |x - \mu|)$.

For training, we use a standard binary cross entropy loss, optionally augmented with a Jacobian regularizer on the Frobenius norm of the network Jacobian scaled by $\lambda > 0$ [31]. As our certified radii in Theorem 3.1 vary inversely to the norm of the Jacobian, this regularization helps boost our certificates at a minimal loss in clean accuracy. We choose $\lambda = 0.0075$ for CIFAR-10, $\lambda = 0.075$ for Malimg and $\lambda = 0.01$ for MNIST and Fashion-MNIST. Further ablation tests studying the impact of regularization are reported in Appendix G. All feature-convex networks are trained using SGD with a learning rate of $0.001$, momentum $0.9$, and exponential learning rate decay with $\gamma = 0.99$.

Table 3: Convex ConvNet architecture parameters. $C_1$ denotes the first convolution, with $C_{2,\ldots}$ denoting all subsequent convolutions. The "Features" column denotes the number of output features of $C_1$, which is held fixed across $C_{2,\ldots}$. The "Pool" column refers to the size of the final MaxPool window before the linear readout layer. The MNIST and Malimg architectures are simple multilayer perceptrons and are therefore not listed here.

| Dataset | Features | Depth | $C_1$ size | $C_1$ stride | $C_1$ dilation | $C_{2,\ldots}$ size | Pool |
|---|---|---|---|---|---|---|---|
| Fashion-MNIST | 4 | 3 | 5 | 1 | 1 | 3 | 1 |
| CIFAR-10 | 16 | 5 | 11 | 1 | 1 | 3 | 1 |

### E.4  Class Accuracy Balancing

As discussed in Section 1.1, a balanced class 1 and class 2 test accuracy is essential for a fair comparison of different methods. For methods where the output logits can be directly balanced, this is easily accomplished by computing the ROC curve and choosing the threshold that minimizes $|\text{TPR} - (1 - \text{FPR})|$. This includes both our feature-convex classifiers with one output logit and the Cayley orthogonalization and $\ell_\infty$-Net architectures with two output logits.

Randomized smoothing classifiers are more challenging as the relationship between the base classifier threshold and the smoothed classifier prediction is indirect. We address this using a binary search balancing procedure. Namely, on each iteration, the classifier's prediction routine is executed over the test dataset and the "error" between the class 1 accuracy and the class 2 accuracy is computed. The sign of the error then provides the binary signal for whether the threshold should be shifted higher or lower in the standard binary search implementation. This procedure is continued until the error drops below $1\%$.

## F  $\ell_2$- and $\ell_\infty$-Certified Radii

This section reports the counterpart to Figure 2 for the $\ell_2$- and $\ell_\infty$-norms. Across all experiments, we attain substantial $\ell_2$- and $\ell_\infty$-radii without relying on computationally expensive sampling schemes or nondeterminism. Methods that certify to another norm $\|\cdot\|_p$ are converted to $\ell_q$-radii at a factor of 1 if $p > q$ or $d^{1/p-1/q}$ otherwise.

Certified $\ell_2$-radii are reported in Figure 9. Our $\ell_2$-radii are moderate, generally slightly smaller than those produced by Gaussian randomized smoothing.

Certified $\ell_\infty$-radii are reported in Figure 10. For the MNIST 3-8 experiment, the $\ell_\infty$-distance nets produce exceptional certified radii. Likewise, the $\ell_\infty$-distance net certificates are dominant for the Fashion-MNIST dataset, despite achieving slightly inferior clean accuracy. We note however that the applicability of $\ell_\infty$-distance nets for sophisticated vision tasks is uncertain as the method is unable to achieve better-than-random performance for CIFAR-10 cats-dogs (Section E.2). Our method is comparable to randomized-smoothing and $\alpha, \beta$-CROWN in all $\ell_\infty$ experiments.

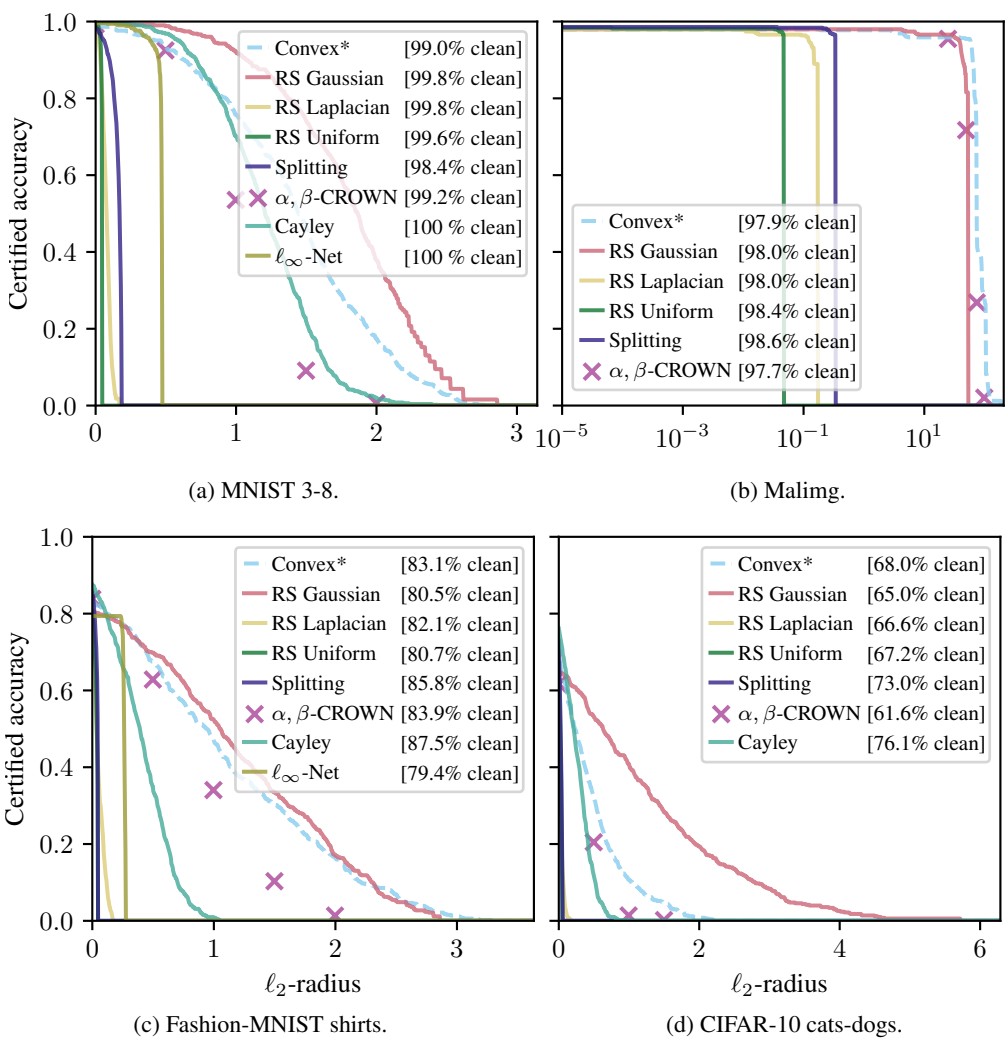

(a) MNIST 3-8.

(b) Malimg.

(c) Fashion-MNIST shirts.

(d) CIFAR-10 cats-dogs.

Figure 9: Class 1 certified radii curves for the $\ell_2$-norm.

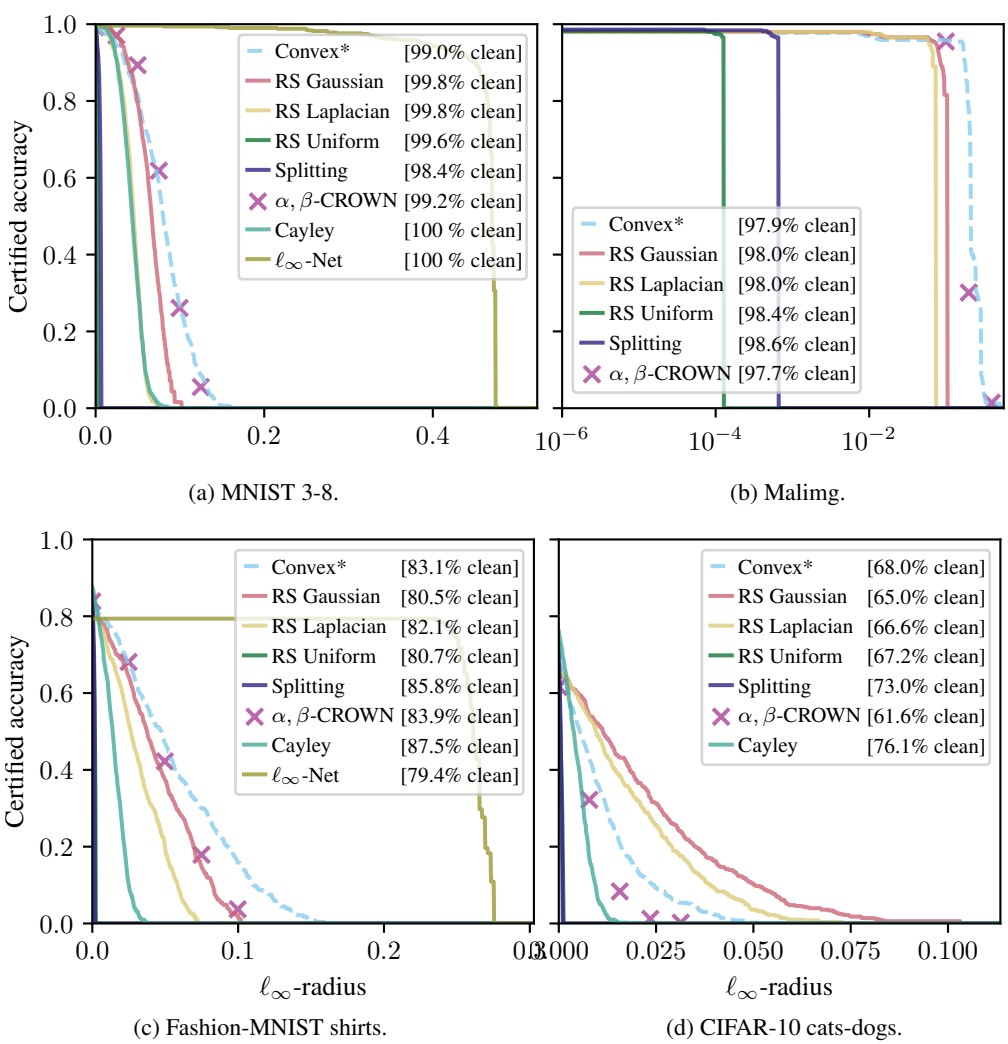

(a) MNIST 3-8.

(b) Malimg.

(c) Fashion-MNIST shirts.

(d) CIFAR-10 cats-dogs.

Figure 10: Class 1 certified radii curves for the $\ell_\infty$-norm.

# G  Ablation Tests

We conduct a series of ablation tests on the CIFAR-10 cats-dogs dataset, examining the impact of regularization, feature maps, and data augmentation.

## G.1  Regularization

Figure 11 examines the impact of Jacobian regularization over a range of regularization scaling factors $\lambda$, with $\lambda = 0$ corresponding to no regularization. As is typical, we see a tradeoff between clean accuracy and certified radii. Further increases in $\lambda$ yield minimal additional benefit.

## G.2  Feature Map

In this section, we investigate the importance of the feature map $\varphi$. Figure 12 compares our standard feature-convex classifier with $\varphi(x) = (x - \mu, |x - \mu|)$ against an equivalent architecture with $\varphi = \mathrm{Id}$. Note that the initial layer in the convex ConvNet is a BatchNorm, so even with $\varphi = \mathrm{Id}$, features still get normalized before being passed into the convolutional architecture. We perform this experiment across both the standard cats-dogs experiment (cats are certified) in the main text and the reverse dogs-cats experiment (dogs are certified).

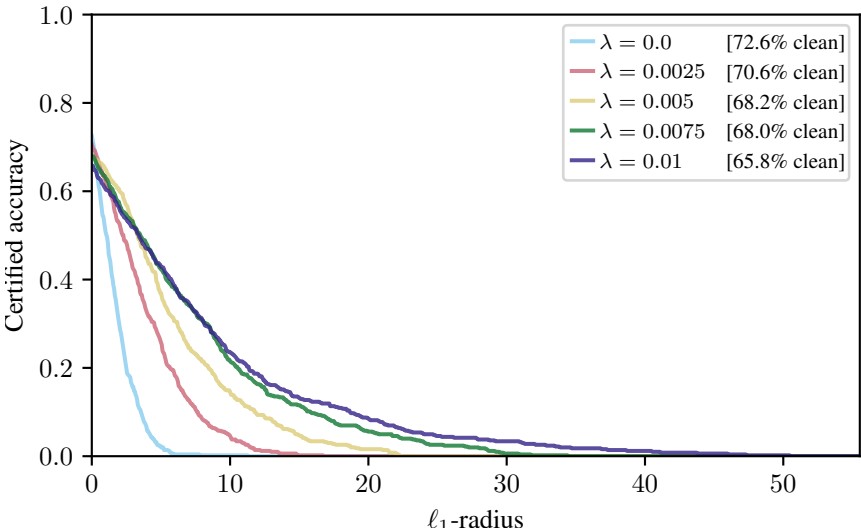

Figure 11: Impact of the Jacobian regularization parameter $\lambda$ on CIFAR-10 cats-dogs classification.

As expected, the clean accuracies for both datasets are lower for $\varphi = \mathrm{Id}$, while the certified radii are generally larger due to the Lipschitz scaling factor in Theorem 3.1. Interestingly, while the standard $\varphi$ produces comparable performance in both experiments, the identity feature map classifier is more effective in the dogs-cats experiment, achieving around 7% greater clean accuracy. This reflects the observation that convex separability is an asymmetric condition and suggests that feature maps can mitigate this concern.

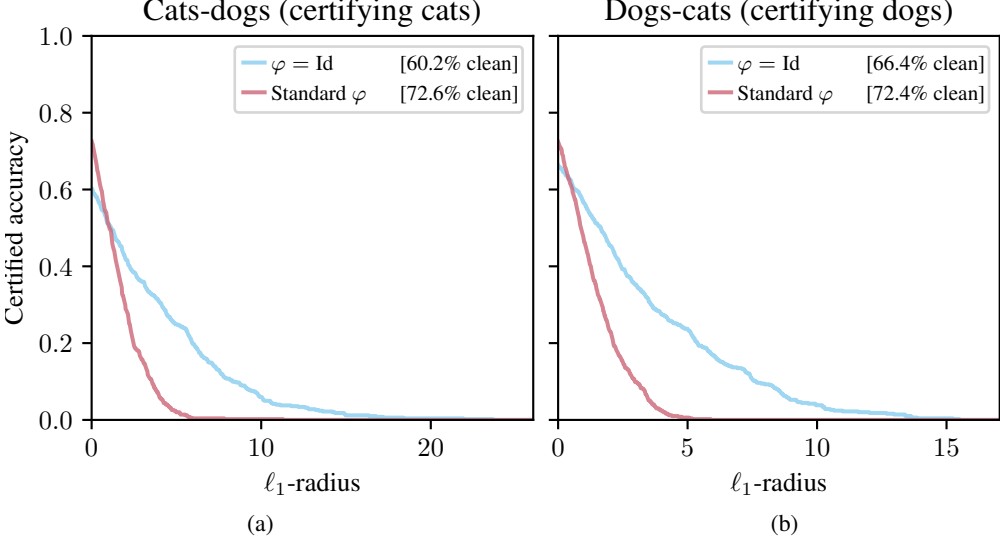

Figure 12: (a) Certification performance with cats as class 1 and dogs as class 2. (b) Certification performance with dogs as class 1 and cats as class 2.

### G.3 Unaugmented Accuracies

Table 4 summarizes the experimental counterpart to Section 3.2. Namely, Fact 3.8 proves that there exists an input-convex classifier ($\varphi = \mathrm{Id}$) that achieves perfect training accuracy on the CIFAR-10 cats-dogs dataset with no dataset augmentations (random crops, flips, etc.). Our practical experiments

are far from achieving this theoretical guarantee, with just 73.4% accuracy for cats-dogs and 77.2% for dogs-cats. Improving the practical performance of input-convex classifiers to match their theoretical capacity is an exciting area of future research.

Table 4: CIFAR-10 accuracies with no feature augmentation ($\varphi = \mathrm{Id}$) and no input augmentation.

| Class 1-class 2 data | Training accuracy | Test accuracy (balanced) |
|---|---|---|
| Cats-dogs | 73.4% | 57.3% |
| Dogs-cats | 77.2% | 63.9% |

## H  MNIST Classes Sweep

For our comparison experiments, we select a specific challenging MNIST class pair (3 versus 8). For completeness, this section includes certification results for our method over all combinations of class pairs in MNIST. As this involves training models over 90 combinations, we lower the number of epochs from 60 to 10, maintaining all other architectural details described in Appendix E.

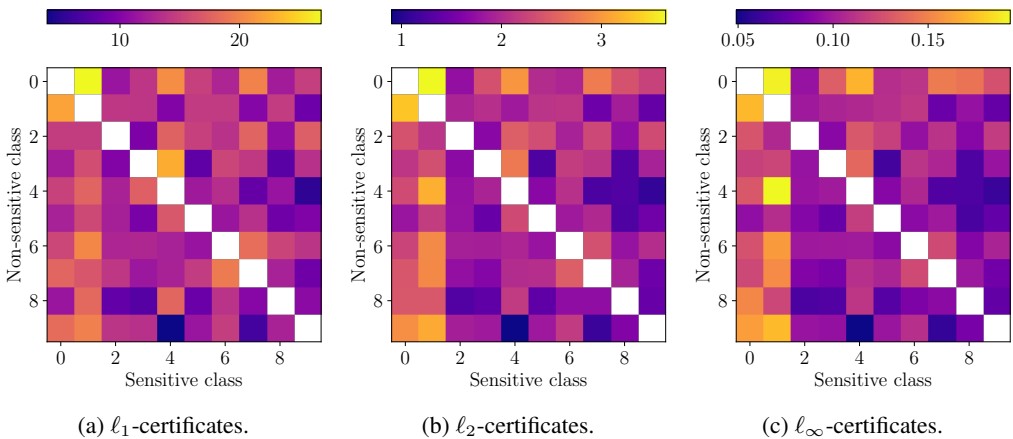

(a) $\ell_1$-certificates.  (b) $\ell_2$-certificates.  (c) $\ell_\infty$-certificates.

Figure 13: Plotting the median certified radii for the MNIST feature-convex architecture over a range of class combinations. The horizontal axis is the class being certified. The MNIST 3-8 experiment considered throughout therefore corresponds to the cell $(3, 8)$ in each plot.

Our certified radii naturally scale with the complexity of the classification problem. As expected, 3 and 8 are among the most challenging digits to distinguish, along with 2-8, 5-8, 4-9, and 7-9. Particularly easy combinations to classify typically include 0 or 1.

The certification performance is remarkably symmetric across the diagonal despite the asymmetry in our convex architectures. In other words, when classifying between digits $i$ and $j$, if a convex classifier exists which generates strong certificates for $i$, then we can generally train an asymmetric classifier that generates strong certificates for $j$. A few exceptions to this can be seen in Figure 13; the most notable are the 1-9 versus 9-1 pairs and the 4-8 versus 8-4 pairs. A deeper understanding of how class characteristics affect asymmetric certification is an exciting avenue of future research.

## I  Randomized Smoothing Noise Level Sweeps

In this section, we reproduce the performance randomized smoothing classifiers under different noise distributions for a range of noise parameters $\sigma$. Namely, we sweep over multiples of base values of $\sigma$ reported in the subcaptions of Figures 14, 15, and 16. The base values of $\sigma$ were set to $\sigma = 0.75$ for the MNIST 3-8, Fashion-MNIST, and CIFAR-10 cats-dogs experiments. For the higher-resolution Malimg experiment, we increase the base noise to $\sigma = 3.5$, matching the highest noise level examined in Levine and Feizi [40]. The epochs used for training were similarly scaled by $n$, starting from the base values provided in Section E, with the exception of the CIFAR-10 base epochs being increased to 600 epochs.

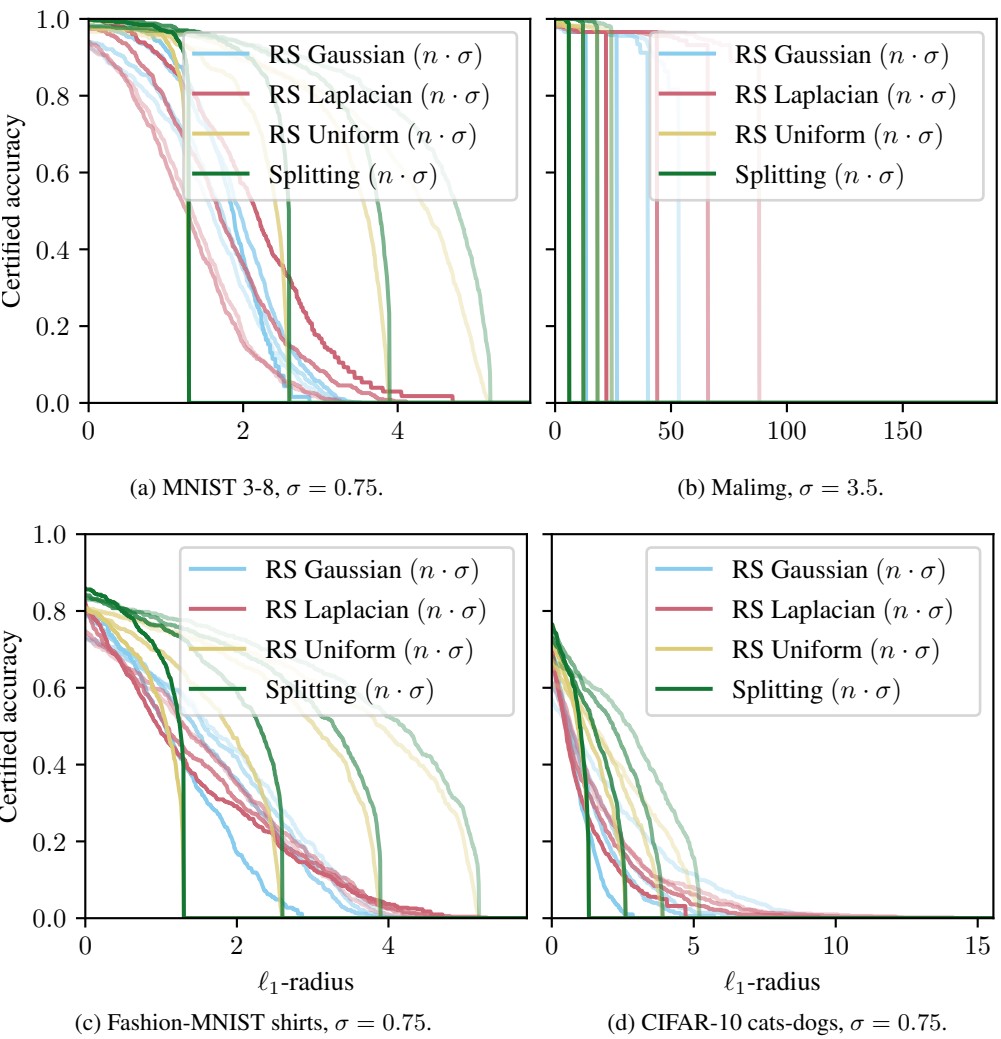

(a) MNIST 3-8, $\sigma = 0.75$.

(b) Malimg, $\sigma = 3.5$.

(c) Fashion-MNIST shirts, $\sigma = 0.75$.

(d) CIFAR-10 cats-dogs, $\sigma = 0.75$.

Figure 14: Randomized smoothing certified radii sweeps for the $\ell_1$-norm. Line shade indicates value of the integer noise multiplier $n$, with $n$ ranging from $1$ (darkest line) to $4$ (lightest line).

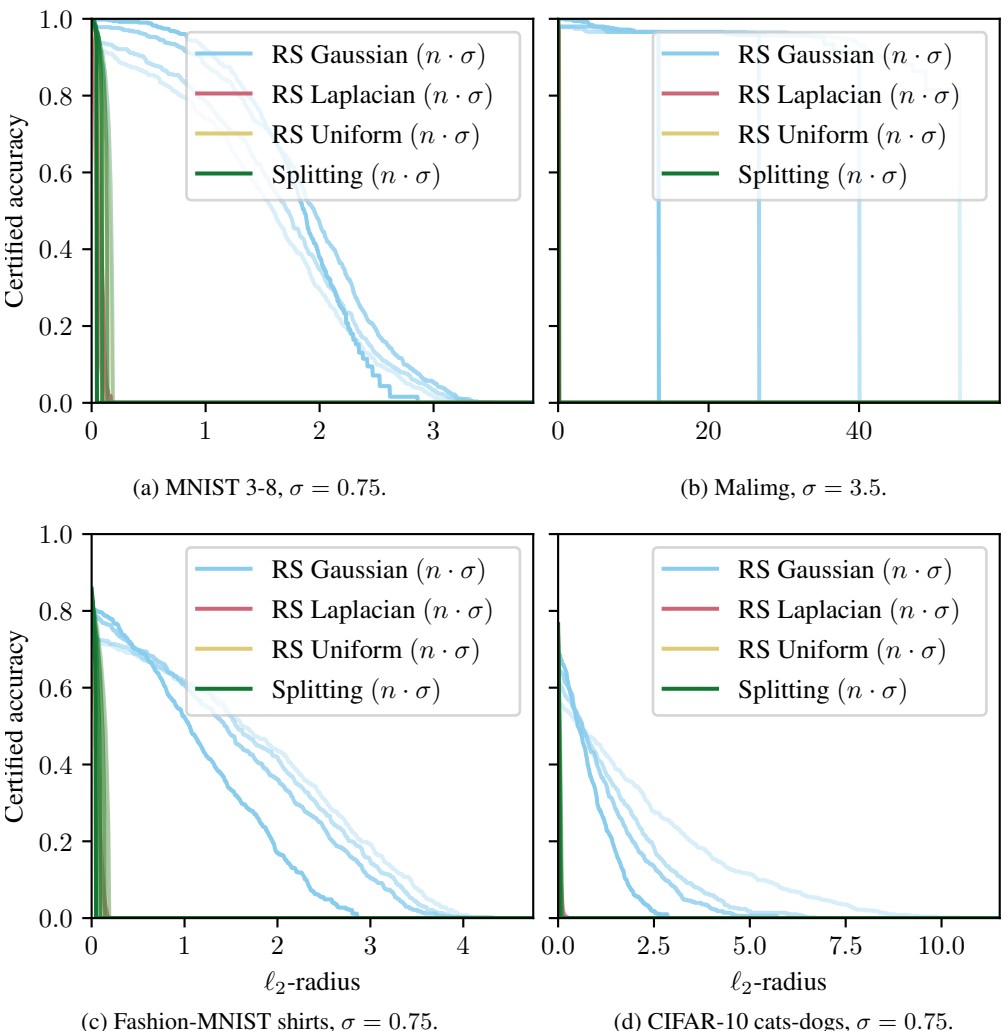

(a) MNIST 3-8, $\sigma = 0.75$.

(b) Malimg, $\sigma = 3.5$.

(c) Fashion-MNIST shirts, $\sigma = 0.75$.

(d) CIFAR-10 cats-dogs, $\sigma = 0.75$.

Figure 15: Randomized smoothing certified radii sweeps for the $\ell_2$-norm. Line shade indicates value of the integer noise multiplier $n$, with $n$ ranging from $1$ (darkest line) to $4$ (lightest line). For higher-dimensional inputs (Malimg and CIFAR-10) methods which certify to a different norm and convert are uncompetitive.

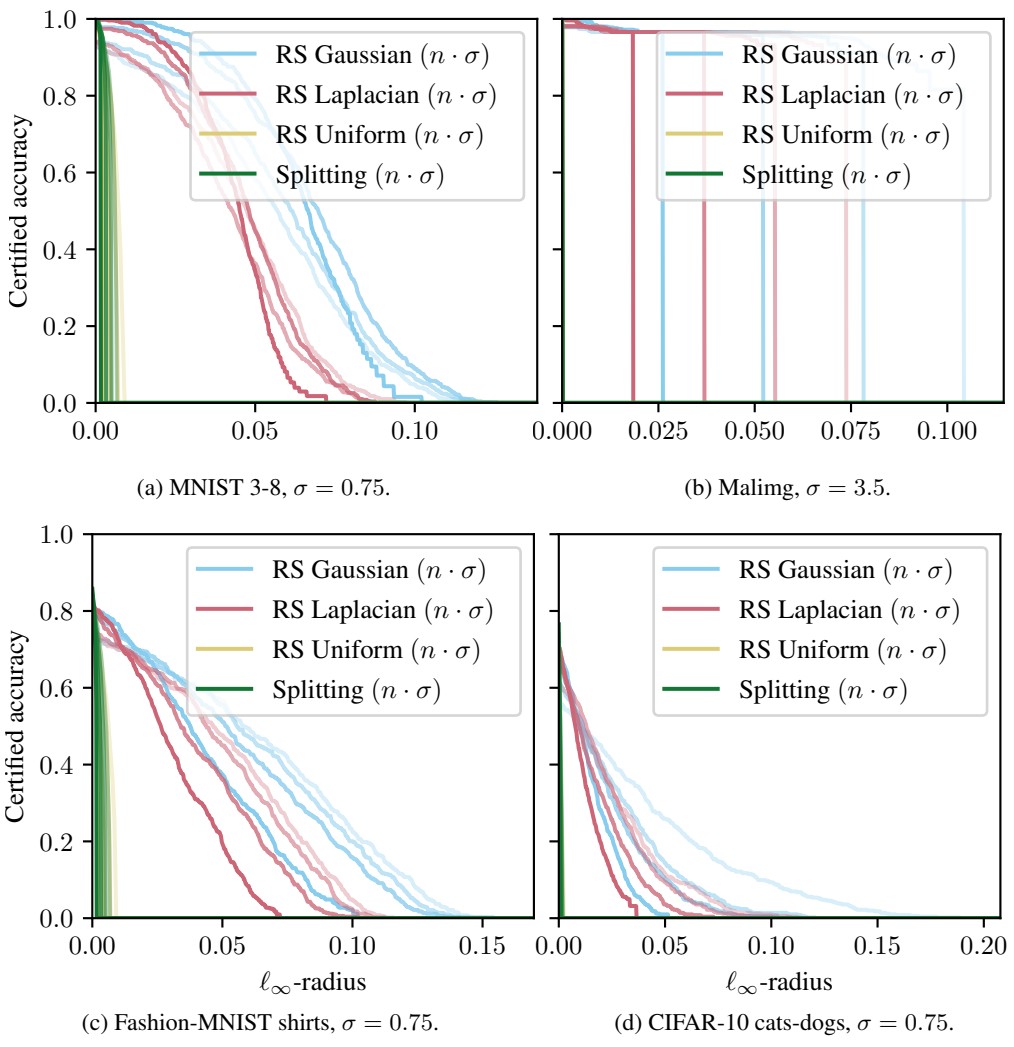

(a) MNIST 3-8, $\sigma = 0.75$.

(b) Malimg, $\sigma = 3.5$.

(c) Fashion-MNIST shirts, $\sigma = 0.75$.

(d) CIFAR-10 cats-dogs, $\sigma = 0.75$.

Figure 16: Randomized smoothing certified radii sweeps for the $\ell_\infty$-norm. Line shade indicates value of the integer noise multiplier $n$, with $n$ ranging from $1$ (darkest line) to $4$ (lightest line).

