# OpenReview forum: "Asymmetric Certified Robustness via Feature-Convex Neural Networks"
_NeurIPS.cc/2023/Conference — NeurIPS 2023 poster_

### Official Review · Reviewer_g6eM · 2023-06-23

**Soundness:** 3 good
**Presentation:** 4 excellent
**Contribution:** 2 fair
**Rating:** 6
**Confidence:** 4

**Summary:**

This paper is based on the following elegant observation: Consider the case of binary classification in which we learn a function $f: \mathbb R^e\to \mathbb R$ and classify a point based on thresh-holding $f$ at 0. Assume $f(x)=g(\phi(x))$ where $f:\mathbb R^m \to \mathbb R$ is convex and $\phi$ is Lipschitz. Then using the Lipschitz consonant of $\phi$ and the fact that $g(y)\geq g(\phi(x))+\nabla g(\phi(x))\cdot (h-\phi(x))$ for any subgradient of $g$, one can easily compute a robustness certification for $f$. The authors then argue that this paradigm is fairly applicable to adversarial binary classification based on the following observations: 1) If $m$ is larger than the number of data points and $\phi$ is the identity, then there is a convex function achieving perfect accuracy. 2) There are many adversarial classification tasks involving asymmetric binary classification such as malware detection and spam detection

**Strengths:**

- computation of a robust certificate for this method is much faster than competing methods and achieves comparable performance
- formalizes the fact that in many real world applications, adversarial classification is an asymmetric problem
- exposition is excellent and bibliography is very thorough

**Weaknesses:**

- Binary classification problems are often handled by SVMs, for which a robust certification is easy to compute. These models are simpler and easier to train than neural nets. Furthermore, one expects some robustness to arise from these models as they minimize the maximum margin. This paper does not compare accuracy or robustness with SVMs. For instance, due to Corrollary 3.8, I would expect SVMs to perform well on the CIFAR 10 example. Can you find an example where your method does better than an SVM in terms of either accuracy or robustness?
- Consider $\hat g$ as defined in definition 2.2 If $b_2,\ldots b_\ell\geq 0$, the last $\ell-1$ layers reduce to a linear function. This observation suggests that depth for convex neural nets has fairly different effects than depth for standard neural nets. This observation further suggests that to achieve an expression $\hat g$, $\ell$ would need to be quite large. Can you elaborate on your choice of $\ell$?
- There are many instances for which the dimension of the dataset $d$ is less than the number of data points. One would expect that using a non-identity $\phi$ in these cases would be advantageous. The paper does not discuss how to choose $\phi$ in this case
- Comment: I would call "Corollary 3.8" a "Fact" rather than a "Corollary". Optimization in finite precision arithmetic always has errors, which is weaker than a formal proof.

**Questions:**

See the questions in the first two bullets of "Weaknesses". A convincing response to the first bullet could significantly change my opinion on the rating of this paper.

**Limitations:**

See the first two bullets under "Weaknesses"

---

> ### Author Rebuttal · Authors · 2023-08-05
>
> Thank you for your kind complement on the exposition of our paper. We appreciate your constructive suggestions, which we address below.
>
> 1. "Can you find an example where your method does better than an SVM in terms of either accuracy or robustness?":
>
> Certainly, we have shared the relevant script with the AC through an anonymized link (as per the rebuttal guidelines). For the CIFAR-10 task that you mentioned, a linear SVM only achieves a 54.8% clean accuracy, compared to 68% for our method (with regularization). We also experimented with adding the same Lipschitz-continuous feature map that we used for our architecture (the concatenation $x\mapsto (x,|x|)$, after shifting the images to be in the $[-0.5, 0.5]$ range). This only marginally improved the clean accuracy of the SVM to 55.6%. Certification performance is also roughly an order of magnitude worse than our method across all norms---e.g., the maximum $\ell_1$-norm certified radius was just 2.93, compared to over 30 for our method. This reflects the fact that convex classifiers are significantly more flexibile than linear classifiers, while retaining fast certification computation by using linear underapproximators. Note that Corollary 3.8 applies to the class of input-convex ReLU neural networks; it does not imply that an SVM can achieve perfect training accuracy, since SVMs constitute only a small subset of all possible convex classifiers. For an SVM to achieve perfect training accuracy, the dataset would need to be linearly separable, which is a much stronger condition than the convex separability in both directions (cats-dogs and dogs-cats) that we show (e.g., consider class 1 points at $(-1,0),(1,0)\in\mathbb{R}^d$ and class 2 points at $(0,-1),(0,1)\in\mathbb{R}^2$, which are convexly separable in both directions but not linearly separable).
>
> Here are the clean accuracies and maximum certified radii for the SVM on the CIFAR-10 cats-dogs task, outputted from the script:
>
> **With no feature map:**
>
> Clean accuracy: 0.548
>
> l1: 2.930618825346973
>
> l2: 0.2083568160433327
>
> linf: 0.004723921349099762
>
> **With the $(x, |x|)$ feature map:**
>
> Clean accuracy: 0.556
>
> l1: 1.4921445716893802
>
> l2: 0.11696300733917692
>
> linf: 0.002664742804160659
>
> 2. "Consider $\hat{g}$ as defined in definition 2.2...":
>
> The biases $b^{(l)}$ are not constrained to be nonnegative, and even if they are nonnegative, the $l$th layer may still be nonlinear in the "passthrough" $x^{(0)}$, since $C^{(l)}x^{(0)}$ is fed into the activation with $C^{(l)}$ having possibly negative elements. This nonlinearity then propagates into all subsequent layers.
>
> We selected our layer count $L$ empirically based on certification performance, finding that relatively shallow networks tend to suffice (i.e., $5$ layers for CIFAR-10 cats-dogs). We presume that this is tied to the documented tendency for ICNNs to avoid overfitting (see [60] in our manuscript), although a more thorough investigation of this phenomenom is better suited for future work.
>
> 3. "The paper does not discuss how to choose $\phi$ in this case":
>
> The relationship between dataset dimensionality and the number of datapoints is quite complex, and our feeling is that the characteristics of the dataset distribution are practically more important than its dimension. For example, many noncomplex datasets in lower dimensions might be well classified with a simple classifier, and don't need any feature map (i.e., the MNIST 3-8 dataset). On the other hand, the higher dimensional CIFAR cats-dogs dataset benefited significantly from a feature map (see appendix G.2). While our Theorem 3.9 does begin to look at the interplay of dimension and separability, a more empirical investigation of this phenomenon and appropriate feature maps is better suited for future work.
>
> 4. "Optimization in finite precision arithmetic always has errors, which is weaker than a formal proof":
>
> Your point on finite precision is a good one. Thus, we have re-labeled the result as a "Fact," per your suggestion.
>
> Q1. "A convincing response to the first bullet could significantly change my opinion on the rating...":
>
> Thank you for your openness. We have addressed each of your raised points, and in particular demonstrated the superiority of our method over SVMs, as per your first bullet point. We hope that you find our updates sufficient to significantly change your rating, and appreciate your help in enhancing the paper.

---

> > ### Comment · Reviewer_g6eM · 2023-08-10
> > **Nice SVM results!**
> >
> > 1. Your comparison with SVMs is compelling. I have updated my score.
> > At the same time, the clean accuracy of your method is much closer to an SVM than a neural net.
> >
> > Sorry, my mistake on Corollary 3.8. Consider including that example of points in $\mathbb R^2$ that are convexly separable but not linearly separable in your paper.
> >
> > 2. Great! I think you should also include this discussion in you paper. This has intuitively helped me understand the sources of non-linearity in your nets.

---

### Official Review · Reviewer_44oN · 2023-07-03

**Soundness:** 4 excellent
**Presentation:** 1 poor
**Contribution:** 4 excellent
**Rating:** 6
**Confidence:** 4

**Summary:**

This paper tackles the asymmetric nature of classification and adversarial attacks intrinsic in most real-world scenarios that have a live and motivated attacker: that attacks are unidirectional, and proposes a general-purpose technique for specifying a certifiably robust defense in such scenarios. This is done by using an Input Convex Neural Network (ICNN) to guarantee the convexity of the function $g(x)$, and then double-backprop is applied to get the gradient of the ICNN with respect to its input to compute the certificate radius.

-------

I think my concerns are mostly addressed. I'm waffling between 6/7, so at least updating my minimum confidence.

**Strengths:**

1. Valuable theoretical contribution to how adversarial machine learning works in real-life, over unmotivated fears about a computer-vision boogie man.
2. Comparisons against many alternative certifiable defenses, with significant improvement in some cases.
3. Should be faster, but such results are obscured.

**Weaknesses:**

1. The experimental section is woefully presented and obscures the nature of the benefit, abdicating the setup/explanation to the appendix is not acceptable in my opinion.
2. The mathematical presentation (additionally not helped by the NeurIPS template) is hard to follow, e.g. the setup for the reader in section 1.3 is crammed into a tightly packed paragraph to hit the page limit, and the readability suffers.


**Questions:**

1. How much faster is your approach at certifying a radius against each other classifier? This should be in the experiments section to justify the value of the method.
2. Is there a predictive difference in your approach verses other methods (I suspect yes, e.g., randomized smoothing)? See Q1, it should be in Experiments.
3. I read the appendix and it says the "pixels" in malimage are [0, 255] encoded. Wouldn't that mean 10^4/255 = 39 pixels can be altered? It looks like others are right around the one-pixel attack radius in this case.
4. Why not use the neural network approach/data from _SOREL-20M: A large scale benchmark dataset for malicious PE detection_? The MalImage dataset is highly flawed, taking an approach ("malware images") that has been long known to be errant (_Malware Detection by Eating a Whole EXE_) and has significant issues in failing to account for biases that occur in the malware space (_TESSERACT: Eliminating Experimental Bias in Malware Classification across Space and Time_).

**Limitations:**

Limitations are maybe sufficiently addressed, see questions.

---

> ### Author Rebuttal · Authors · 2023-08-05
>
> Thank you for your comments and questions. We have addressed them below, and revised the manuscript accordingly.
>
> 1. "The experimental section...":
>
> As mentioned in our response to Reviewer qP9X, we have revised the Experiments section to satisfy your suggestions. In particular, we have moved the most pertinent details of the experimental setup from Appendix E to Section 4, and we have explicitly defined the keywords in the legends, including the baselines (with citations) as well as the clean accuracy numbers. We believe these edits make the presentation of our experimental results self-contained and much clearer to the reader. Please find the revised section in the general response.
>
> 2. "The mathematical presentation... is hard to follow, e.g.... section 1.3":
>
> We made the following revisions to the mathematical presentation according to your comments and the suggestions from Reviewer qP9X:
>
> a) We have moved Propositions 3.4 and 3.5 (that characterize the decision region geometry) to Appendix C, for two reasons:
>
> i) To more clearly highlight the important theorems and avoid a long sequence of mathematically dense results.
>
> ii) To make room for our experimental details added to Section 4. In place of these propositions, we have modified lines 250--254 with the more easily understood description:
>
> "Although low-dimensional intuition may cause concerns regarding the convex separability of sets of binary-labeled data, we will soon see in Corollary 3.8 and Theorem 3.9 that, even the CIFAR-10 cats and dogs classes are convexly separable, as well as relatively unstructured binary datasets in high dimensions (with high probability). This convex separability of datasets is an highly advantageous property, as input-convex classifiers may always perfectly fit such data, and conversely, input-convex classifiers have interpretable decision regions consisting of a convex set and its complement (the latter of which is not necessarily true for our more general feature-convex architectures with $\varphi\ne\text{Id}$). We formally state and prove these geometric characterizations in Propositions 3.4 and 3.5 of Appendix C."
>
> b) We have updated the first paragraph of Section 3 to clarify the importance of the presented results:
>
> "We present our main theoretical results in this section. First, we derive asymmetric robustness certificates (Theorem 3.1) for our feature-convex classifiers in Section 3.1. Then, in Section 3.2, we introduce the notion of convexly separable sets in order to theoretically characterize the representation power of our classifiers. Our primary representation results give a universal function approximation theorem for our classifiers with $\varphi=\text{Id}$ and ReLU activation functions (Theorem 3.6) and show that such classifiers can perfectly fit convexly separable datasets (Theorem 3.7), including the CIFAR-10 cats-dogs training data (Corollary 3.8). We also show that this strong learning capacity generalizes by proving that feature-convex classifiers can perfectly fit high-dimensional uniformly distributed data with high probability (Theorem 3.9)."
>
> Since the notations introduced in Section 1.3 are used throughout the paper, we find it best to define them up front and in their own short section for reference. We are open to any suggestions you may have for enhancing the readibility of Section 1.3.
>
> Q1. "How much faster is your approach...":
>
> To make the runtime discussion in Section 4 more explicit, we ran a quick benchmark on the CIFAR cats-dogs dataset:
>
> | Method | Certification time (s) |
> | ------ | ------------------ |
> | FCNN | 0.0021 |
> | RS Laplace | 5.95 |
> | RS Gauss | 5.89 |
> | RS Uniform | 5.91 |
> | Splitting | 0.08 |
> | Cayley | 0.052 |
>
> All RS methods use 100,000 samples. $\alpha,\beta$-CROWN is more difficult to compare directly as runtime is computed per property; but generally it is far more computationally expensive and takes on the order of tens of seconds. We also note that the splitting method scales linearly with noise and image size and thus takes on the order of several minutes per sample on Malimg.
>
> Q2. "Is there a predictive difference...":
>
> Yes, the predictions and certificates are fundamentally different in nature between the three randomized smoothing baselines and deterministic methods (including our deterministic method). We have added the following sentence to the Experiments section to remind the reader of this fact and more clearly categorize the different methods in consideration:
>
> "Notice that the three randomized smoothing baselines have fundamentally different predictions and certificates than the deterministic methods (including ours), namely, the predictions are random and the certificates hold only with high probability."
>
> Q3. "Wouldn't that mean 10^4/255 = 39 pixels can be altered?":
>
> We normalize the pixel values to $[0, 1]$. We have added the clarifying sentence "All pixel values are normalized into the interval $[0,1]$." to Section 4 of the manuscript.
>
> Q4. "Why not use ... SOREL-20M...":
>
> The baseline network proposed in SOREL-20M operates on features extracted from the binary. An architecture that certifies robustness to changes in features is difficult to analyze, as the implications for robustness in bit-space (which the adversary can directly manipulate) are unclear. While more sophisticated approaches for classifying directly on bit-space exist (see [25] in our manuscript), we consider these outside the aim of our work. Our primary aim is to introduce a more general framework of certified asymmetric robustness (not just for malware). We used Malimg to give an intuitive practical application that naturally fits within this asymmetric framework; we are making no claims regarding whether the Malimg approach is good or bad for the specific problem of malware classification.

---

> > ### Comment · Reviewer_44oN · 2023-08-10
> >
> > >Q1. "How much faster is your approach...":
> >
> > If you could re-run and give a range of Crown timings for the actual image data used, if it is a matter of scaling to observe RS speedup, please do so. It makes the argument far stronger. I think it is acceptable if the result is measured against a few samples in terms of rebuttal, but you would have plenty of time till camera-ready to get that number on the whole test set.
> >
> > > Malimg to give an intuitive practical application that naturally fits within this asymmetric framework; we are making no claims regarding whether the Malimg approach is good or bad for the specific problem of malware classification.
> >
> > I think a warning to the reader that Malimg is done only for illustration, and that the Malimg approach isn't advised for actual deployment (w/ citation), would satisfy my concern with the rest of the explanation.

---

> > > ### Author Response · Authors · 2023-08-11
> > >
> > > *Crown timings:* As a point of reference, we ran a quick $\alpha,\beta$-CROWN experiment on 10 CIFAR cats-dogs images. Verifying a particular property (norm + epsilon) takes on average 17.48 seconds per sample on our hardware. Note that this is actually understates the complexity of computing the certified radius of a particular sample. $\alpha,\beta$-CROWN only provides a binary true/false signal of whether a particular radius is certified; to find the true largest certified radius for a sample would require an iterative scheme (e.g., binary search). Our method directly outputs the certified radius.
> > >
> > > We agree that a more explicit discussion of runtime would highlight the advantages of our method more clearly. We appreciate the suggestion and will include a runtime experiment on the whole test set in the camera ready version.
> > >
> > > *Malimg:* We are happy to put this remark in the camera ready version.
> > >
> > > Please let us know if you have any remaining concerns or suggestions regarding our work!

---

### Official Review · Reviewer_yvd4 · 2023-07-04

**Soundness:** 3 good
**Presentation:** 3 good
**Contribution:** 3 good
**Rating:** 5
**Confidence:** 4

**Summary:**

This paper considers a specific problem in binary classification task, where one class is recognized as a ‘sensitive’ class, which needs to be certified for robustness. The authors propose a special network called feature-convex neural network, which combines a Lipschitz network and a convex ReLU network. Expressive power of input-convex networks for convexly separable sets are provided. Experiments show that proposed networks achieve better performance on $l_1$ robustness compared to other certified robustness methods.

**Strengths:**

1. The paper is well-written and the problem and proposed method are clear.
2. The proposed problem and method are new.
3. The theoretical results are solid.

**Weaknesses:**

1. The proposed feature-convex network may limited classification power to be practically useful. The lower training accuracy on larger dataset like CIFAR-10 may imply that this architecture will be harder to apply in more complex real-world datasets.

2. The verification of convex separability of CIFAR-10 cats vs dogs is confusing. The authors use convex optimization showing that certain cat image is hard to be recovered from the convex combination of dog images. Is this the sufficient evidence for convex separability? As authors mentioned, input-convex classifier is hard to fit the CIFAR-10 cats vs dogs data, which only achieves $70\%+$ training accuracy. To make convex separability convincing, one may at least design an input-convex network architecture to achieve higher training accuracy (even without generalization ability).

3. The experimental results are a little weak. The proposed feature-convex classifier only shows decent performance on $l_1$ robustness, but is suboptimal in $l_2$ and $l_{\infty}$ robustness. As randomized smoothing is a more general certified method which is not designed for asymmetric binary classification, the suboptimality of feature-convex classifier shows the limited power of the proposed network.

4. A two-layers neural network is used to evaluate the certified power of $\alpha,\beta$-CROWN. This seems to be a weaker baseline, as the conv-small net in $\alpha,\beta$-CROWN uses a four-layer convolutional network [1].

5. The applications mentioned in the paper are mostly for imbalanced dataset (one class has more data than another), but the experiments are mostly for balanced dataset.

[1] S Wang, H Zhang, K Xu, X Lin, S Jana, C J Hsieh, and J Z Kolter. Beta-CROWN: Efficient bound propagation with per-neuron split constraints for neural network robustness verification. In NeurIPS 2021.

**Questions:**

1. When selecting the parameter $\tau$ to shift the classification threshold, why use the balanced accuracy $\alpha_2(\tau)=\alpha_1(\tau)$, but not the commonly-used criterion in binary classification, F1-score or AUROC? Does there exist previous work using balanced accuracy for (asymmetric) binary classification?

2. Can your method be extended to multi-class classification tasks?

3. A very simple Lipschitz feature extractor $g$ is used in the proposed classifier. Can a more complex Lipschitz feature extractor be used to keep both certified robustness and accuracy high?

4. In the proposed application on spam classification, why attackers only attempt to fool the classifier toward the “not-spam” class? The converse attack is more detrimental if the classifier recognizes a not-spam, important email as a spam one.

**Limitations:**

See weakness

---

> ### Author Rebuttal · Authors · 2023-08-05
>
> We greatly appreciate your positive feedback on our paper's presentation, proposed problem/approach, and theory.
>
> 1. "The proposed feature-convex network may limit classification power...":
>
> In practice, we find our clean accuracies to be on par with the state-of-the-art robust classification baseline methods at a comparable level of certification performance (c.f., Figure 2). Corollary 3.8 further shows that our model is theoretically capable of attaining perfect training accuracy on CIFAR cats-dogs. Under standard training, our learned model on CIFAR achieves 73.4% accuracy, emphasizing that there is significant room for improvement. We are thus limited by standard architecture designs and training algorithms when applying them to input-convex models--not the capabilities of ICNNs themselves. We therefore hope that this result motivates future research on new algorithms that are specially tailored to FCNNs/ICNNS.
>
> We have revised the manuscript to more clearly pose this open problem to the readers by adding the following "Open Problem" statement at line 292:
>
> "**Open Problem 3.9.** Train an input-convex ReLU neural network that achieves 100% training accuracy on the unaugmented CIFAR-10 cats-versus-dogs dataset."
>
> 2. "The verification of convex separability of CIFAR-10 cats vs dogs is confusing":
>
> We use convex optimization to verify that, if you choose any cat image, it cannot be represented as a convex combination of the dog images. Hence, the set of cat images (call it, $X_1$) lies completely outside the convex hull of the dog images (call dog images $X_2$ and the convex hull of dog images $X$). This is precisely what it means for cats-dogs to be convexly separable: namely, that $X_2\subseteq X$ and $X_1\subseteq \mathbb{R}^d \setminus X$. This analysis of the dataset is therefore sufficient to conclude convex separability. That our learned ICNN (using standard training) achieves 73.4% training accuracy does not alter the fact that cats-dogs is convexly separable, as the above experiment already verifies the separability. As discussed above, this points to the need for novel ICNN designs and training algorithms to unlock their full potential.
>
> 3. "The experimental results are a little weak":
>
> We emphasize that our approach features several substantial benefits over randomized smoothing. Unlike our deterministic certificates, RS is inherently probabilistic (since it is based on an empirical expectation): there is always a strictly positive probability of a RS classifier producing a prediction that violates its own certificates. Furthermore, randomized smoothing approaches are highly computationally intensive, taking on the order of seconds while our certificates are closed-form and can be computed in milliseconds (~$1000\times$ speedup). As our certificates are generally comparable (for $\ell_2$/$\ell_\infty$-norms) or decidedly superior ($\ell_1$-norm), we consider these advantages to be significant enough to highlight the promise of our method.
>
> 4. "A two-layers neural network is used to evaluate ... $\alpha,\beta$-CROWN":
>
> We used a smaller network for $\alpha,\beta$-CROWN to increase the verification performance of the baseline, as larger networks tended to be more computationally intensive to certify at a particular radius and runtime.
>
> 5. "The applications mentioned in the paper are mostly for imbalanced dataset":
>
> We focus our evaluation primarily on balanced datasets to avoid added analysis complexity, as the work is already quite lengthy. We note that our framework and method apply equally to the balanced and unbalanced scenarios.
>
> Q1. "...why use the balanced accuracy...":
>
> We balance the class accuracies in order to provide a fair comparison of certified accuracy curves across different methods. Otherwise, consider Method A and Method B, where Method A has a superior certified accuracy curve and Method B has a higher clean accuracy for the non-sensitive class; it would be difficult to compare the two methods' certified accuracy curves as it is unclear how much of Method A's advantage came from compromising on non-sensitive class accuracy. Thus, we feel our approach is best for providing directly interpretable certified accuracy curves in the asymmetric setting. Note that in practice a user can adjust the threshold however they see fit--we chose this approach to fascilitate fair and standardized comparisons between methods.
>
> Q2. "Can your method be extended to multi-class...":
>
> Yes. On line 168 in the main paper, we refer to the supplemental material where we propose two ways to extend our method to multi-class settings. Our multi-class generalizations allow for efficient and closed-form asymmetric robustness certificates either for one sensitive class, or one "group" of sensitive classes.
>
> Q3. "Can a more complex Lipschitz feature extractor be used...":
>
> Any feature map can be used so long as it provides a bounded Lipschitz constant. However, more complicated feature maps may come at the expense of a larger Lipschitz constant, resulting in smaller robustness certificates. Therefore, the feature map should be kept as simple as possible to assist with (ideally closed-form) computation of a small Lipschitz constant, but sophisticated enough to make the data convexly separable. Designing and/or learning low-Lipschitz yet high-performing feature maps is left as future work.
>
> Q4. "...why attackers only attempt to fool the classifier toward the not-spam class?":
>
> By definition, a message generated by an attacker is spam, and therefore the only way they can possibly fool the classifier is if they craft a message that is classified as non-spam. Our natural goal is therefore to certify that an adversary cannot "lightly edit" a spam email to make it look to us like non-spam. On the other hand, your proposed situation is not adversarial in nature. An adversary would not craft genuinely important non-spam emails while trying to fool the classifier into thinking that they are spam.

---

> > ### Comment · Reviewer_yvd4 · 2023-08-18
> >
> > Thanks for the explanation.
> > I will keep my score due to the limitation on the training accuracy.

---

### Official Review · Reviewer_qP9X · 2023-07-05

**Soundness:** 2 fair
**Presentation:** 2 fair
**Contribution:** 3 good
**Rating:** 6
**Confidence:** 2

**Summary:**

The paper is about certified robustness of neural networks.  In particular the authors explore the concept that there is typically a one-sided attack that needs to be certified since adversaries have certain goals that only work in certain directions (e.g., classify a spam email as ham).  The authors call this framework "asymmetric robustness certification problem" and approach a solution to this problem using a feature-convex neural network architecture.  The analysis is upon ReLU activation functions where the authors formalize and prove several results that are related to the geometry/convexity of the input data and how this can translate into the existence (not the identification) of a neural network that has perfect accuracy on the dataset.  Of course, it is a neural network at the end of the day and can be applied anywhere and observe its performance.  In this direction, eventually the authors evaluate their method on four different datasets against several baselines.

**After Rebuttal:**
I have read the reviews of others as well as the response by the authors. I am happy with the answers provided by the authors as well as their willingness to improve the presentation of the paper. I will upgrade my presentation score from 1 to 2 and also increase my overall score from borderline accept to weak accept.

**Strengths:**

+ A new approach for certified robustness in a framework that deserves attention.
+ The proposed method is backed up with theoretical guarantees.
+ Experimental results on different datasets show that the method behaves well.

**Weaknesses:**

I think the main issue of the paper is the presentation of the results.  Granted the authors have put a lot of effort in this paper as it can be seen from the full paper.  Nevertheless:
+ The authors use in many situations very long sentences and this makes it hard to follow their work, their descriptions, and ultimately give the appropriate merit to their work.
+ In addition to that, Section 3 and especially Section 3.2, is hard to follow as we have a sequence of definitions, propositions, and theorems that the authors prove as part theoretically justifying their method.
+ Not only that, but it is hard to understand what is important and what is not for the story that the authors want to discuss.
+ In Section 4 where we find the experiments, we have Figure 2 where the authors' method is compared against baseline methods. However, neither in the caption of the figure and the subfigures, nor in the actual text of the paper do we get to see what the other baselines are.  There are some keywords in the legends of the different subfigures, but no explanation.  Most likely RS X means randomized smoothing using noise X (but this is left as an exercise to the reader).  No more information about the rest of the cases is given either, neither are papers cited for these methods that are used as baselines (at least not in that section).  Also, there are some some percentage points next to each method which should present (from the context) that these values correspond to accuracy without any perturbations (and hence the "clean" word appearing next to these numbers) but really, these things need to be spelled out on the paper. One cannot really throw some figures on a paper, with keywords resembling names of certain methods but refuse to explain what these methods really are.  After looking into the appendix, indeed the authors there explain the names of the different methods and cite the relevant work, but this is happening 14 pages after the figure was presented to the reader.  I am sorry, but this is not how papers are read.

I am really torn with this paper and I would like to see the views of the other reviewers. Nevertheless, I appreciate theoretical results and I am leaning more towards acceptance.

**Questions:**

Q1. Are ReLU functions instrumental in the results that you prove? Can these results be extended to other activation functions?

Q2. Apart from the experimental proof that cats and dogs from CIFAR-10 is convexly separable, were you aware of any classification datasets that are convexly separable and could motivate the work that you did?  Of course, this is not to diminish the work that you did; I am just asking.

Q3. At the top of page 4 we see that a neural network is defined as a mapping
$f \colon R^d \times R^n \rightarrow R$.
Can you explain why your input is decomposed to $(x, y)$, with $x\in R^d$ and $y\in R^n$, and why we have R in the output?

---

> ### Author Rebuttal · Authors · 2023-08-05
>
> Thank you for your constructive suggestions. We have addressed all of your comments and questions below, and revised the manuscript accordingly.
>
> 1. "The authors use... long sentences":
>
> We have identified and revised some of our lengthier sentences: "Specifically, we assume..." (line 59); "We characterize..." (line 94); "In contrast..." (line 130); "While high..." (line 286). Please let us know if there were other areas of the text that you found convoluted.
>
> 2/3. "Section 3 and especially Section 3.2, is hard to follow"... "it is hard to understand what is important":
>
> We appreciate your feedback on the framing of our theoretical results. As reviewer 44oN raised similar concerns, please see our response and alterations in point 2 for Reviewer 44oN. If you have other suggestions on how we might adjust presentation and/or language to clarify our main contributions to the reader, we would appreciate hearing them.
>
> 4. "In Section 4... we [do not] get to see what the other baselines are":
>
> We have adjusted the experiments section according to each of your points brought up. In particular, we have moved the most pertinent details of the experiments from Appendix E to Section 4, and we have explicitly defined the keywords in the legends, including the baselines (with citations) as well as the clean accuracy numbers. We believe that, in following your suggestions, these edits make the presentation of our experimental results much clearer to the reader. Please find the revised text in the general response.
>
>
> Q1. "Are ReLU functions instrumental in the results that you prove?":
>
> Only Theorems 3.6, 3.7, and the second part of Theorem 3.9 rely on ReLU activation functions, since we prove our uniform approximation theorem (Theorem 3.6) in terms of ReLU activations. In particular, our primary robustness certificate (Theorem 3.1) holds for general feature-convex models, which may be constructed using any convex nondecreasing activation functions. Notice also that the primary result of Theorem 3.9 (the fact that high-dimensional uniform datasets are convexly separable with high probability) is unrelated to architecture choice.
>
> Q2. "...were you aware of any classification datasets that are convexly separable?":
>
> Yes. As mentioned in Appendix D, "Yousefzadeh [74] and Balestriero et al. [9] showed a related empirical result for CIFAR-10, namely, that no test set image can be reconstructed as a convex combination of training set images." It is important to note that their result does not show the convex separability of two classes of training images, and therefore does not allow one to conclude that a feature-convex classifier can achieve perfect training accuracy. Therefore, our experiment on the cats-dogs training dataset is required in order for us to come to the conclusion of Corollary 3.8.
>
> Q3. "Can you explain why your input is decomposed...":
>
> The work being cited there (Amos et al. [2]) proposed input-convex neural networks as a means to build a model $f(x,y)$ that is convex in the variable $y$, but possibly nonconvex in $x$. They use such a variable decomposition for the specific purposes of optimization-based inference with a real-valued output, i.e., mapping an input $x$ to an ouput defined by the convex optimization problem $\arg\min_{y} f(x,y) \in \mathbb{R}$. Since, in their application of optimization-based inference, one of the inputs to $f$ is being "optimized out," they need to include an auxiliary/decomposed input so that there is still an input $x$ to be fed to $\arg\min_y f(x,y)$. On the other hand, our work, as well as many other works that use input-convexity (e.g., [16,17,48,75,79] in our manuscript), are not interested in optimization-based inference, but rather seek to exploit input-convexity for other reasons, so there is no part of the model input being "optimized out." In our case, we exploit input convexity for purposes of model robustness. Therefore, our work (and the others listed) have no need for an auxiliary input or an input decomposition. That is, we simply use models $g(x)$ that are input-convex in the entire variable $x$.

---

> > ### Comment · Reviewer_qP9X · 2023-08-15
> > **Thank you for the response**
> >
> > I have read the reviews of others as well as the response by the authors.  I am happy with the answers provided by the authors as well as their willingness to improve the presentation of the paper.  I will upgrade my presentation score from 1 to 2 and also increase my overall score from borderline accept to weak accept.

---

> > > ### Author Response · Authors · 2023-08-15
> > >
> > > We thank the reviewer for their time and constructive comments, as well as for taking our updates into consideration for their revised score.

---

### Author Rebuttal · Authors · 2023-08-05

We sincerely thank the Reviewers for their insightful comments and valuable suggestions. We are happy to see that 3/4 reviews are generally positive on our work, with primarily presentation-focused concerns that we address below. Reviewer g6eM has indicated their willingness to update their score with a convincing SVM comparison, which we have also provided.

Please see our individual responses to each Reviewer, where we address each point raised and highlight the corresponding revisions we have made to the manuscript. Here, we briefly describe the main revisions:

1. Mathematical presentation: we have edited and added descriptions in Section 3 to ensure the theoretical results are easily understood by the reader, and to more clearly illustrate the importance of such results and how they play into the overall story of the paper. In doing so, we moved Propositions 3.4 and 3.5 to Appendix C, and added in their place a more easily readable description of what they say.

2. Experiments: we have moved the most pertinent details of the experiments from Appendix E to Section 4, explicitly describing the datasets, baselines (with citations), and our model's architecture, in addition to defining the keywords in the legends and the clean accuracy numbers reported in Figure 2. We believe that, in following the Reviewers' suggestions, these edits make the presentation and contextualization of our experimental results much clearer to the reader and aid in justifying our work.

_Experimental section addition._

"This section compares our feature-convex classifiers against a variety of state-of-the-art baselines in the asymmetric setting. Before discussing the results, we briefly describe the datasets, baselines, and architectures used. For a more in-depth description and hyperparameter details, see Appendix E.

**Datasets.** We evaluate our approach using four datasets. First, we consider distinguishing between $28\times 28$ greyscale MNIST digits $3$ and $8$ [37], which are generally more visually similar and challenging to distinguish than other digit pairs. Next, we consider identifying malware from the "Allaple.A" class in the Malimg dataset of $512\times 512$ bytewise encodings of malware [51]. We then consider distinguishing between shirts and T-shirts in the Fashion-MNIST dataset of $28\times 28$ greyscale images [70], which tend to be the hardest classes to distinguish [33]. Finally, we consider the $32\times 32$ RGB CIFAR-10 cat and dog images since they are again relatively difficult to distinguish [26,44,30]. The latter two datasets can be considered as our more challenging settings. All pixel values are normalized into the interval $[0,1]$.

**Baseline Methods.** We consider several state-of-the-art randomized and deterministic baselines. For all datasets, we evaluate the randomized smoothing certificates of [72] for the Gaussian, Laplacian, and uniform distributions trained with noise augmentation (denoted RS Gaussian, RS Laplacian, and RS Uniform, respectively), as well as the deterministic bound propagation framework $\alpha,\beta$-CROWN [66], which is scatter plotted since certification is only reported as a binary answer at a given radius. We also evaluate, when applicable, deterministic certified methods for each norm ball. These include the splitting-noise $\ell_1$-certificates from [40] (denoted Splitting), the orthogonality-based $\ell_2$-certificates from [63] (denoted Cayley), and the $\ell_{\infty}$-distance-based $\ell_{\infty}$-certificates from [77] (denoted $\ell_\infty$-Net). The last two deterministic methods are not evaluated on the large-scale Malimg dataset due to their prohibitive runtime. Furthermore, the $\ell_{\infty}$-Net was unable to significantly outperform a random classifier on the CIFAR-10 cats-dogs dataset, and is therefore only included in the MNIST 3-8 and Fashion-MNIST shirts experiments. Notice that the three randomized smoothing baselines have fundamentally different predictions and certificates than the deterministic methods (including ours), namely, the predictions are random and the certificates hold only with high probability.

**Feature-Convex Architecture.** Our simple experiments (MNIST 3-8 and Malimg) require no feature map to achieve high accuracy ($\varphi=\text{Id}$). The Fashion-MNIST shirts dataset also benefited minimally from the feature map inclusion. For the CIFAR-10 cats-dogs task, we let our feature map be the concatenation $\varphi(x)=(x-\mu,|x-\mu|)$, as motivated by Appendix B, where $\mu$ is the channel-wise dataset mean (e.g., size $3$ for an RGB image) broadcasted to the appropriate dimensions. Our MNIST 3-8 and Malimg architecture then consists of a simple two-hidden-layer input-convex multilayer perceptron with $(n_1,n_2)=(200,50)$ hidden features, ReLU nonlinearities, and passthrough weights. For the Fashion-MNIST shirts (CIFAR-10 cats-dogs, resp.) dataset, we use a convex ConvNet architecture consisting of $3$ ($5$, resp.) convolutional, BatchNorm, and ReLU layers. All models are trained using SGD on the standard binary cross entropy loss with Jacobian regularization, and clean accuracies are balanced as described in Section 1.1 and Appendix E.4 to ensure a fair comparison of different robustness certificates.

**Results and Discussion.** Experimental results for $\ell_1$-norm certification are reported in Figure 2, where our feature-convex classifier radii, denoted by Convex*, are similar or better than all other baselines across all datasets. Also reported is each method's clean test accuracy without any attacks, denoted by "clean."..."

---

### Decision · Program_Chairs · 2023-09-21

**Decision:**

Accept (poster)

**Comment:**

Dear Authors,

Thank you for your valuable contribution to Neurips and the ML community. Your submitted paper has undergone a rigorous review process, and I have carefully read feedback provided by the reviewers and considered the author rebuttal in detail.

This paper is focused on constructing certifiably robust neural networks. The authors employ input convex neural networks and Lipschitz continuity to design robust models. They propose applying double-backprop to get the gradient of the network with respect to its input to compute the robustness radius. The reviewers all agree that the introduced method is novel and interesting, the numerical results are good and writing of the paper is high-quality.

Given this positive assessment, I am willing to recommend the acceptance of your paper for publication.

I would like to remind you to carefully review the reviewer feedback and the resulting discussion. While most reviews were positive, the reviewers have offered valuable suggestions that can further strengthen the quality of the paper. Please take another careful look a the 'weaknesses' section of each reviewer comment. Please also review the discussions with Reviewer yvd4, whose review was the most critical. I encourage you to use the feedback to make any necessary improvements and refinements before submitting the final version of your paper.

Once again, thank you for submitting your work to Neurips.

Best,
Area Chair